# HELIOS: Hierarchical Exploration for Language-grounded Interaction in Open Scenes

## Abstract

Language-specified mobile manipulation tasks in novel environments simultaneously face challenges interacting with a scene which is only partially observed, grounding semantic information from language instructions to the partially observed scene, and actively updating knowledge of the scene with new observations. To address these challenges, we propose HELIOS, a hierarchical scene representation and associated search objective to perform language specified pick and place mobile manipulation tasks. We construct 2D maps containing the relevant semantic and occupancy information for navigation while simultaneously actively constructing 3D Gaussian representations of task-relevant objects. We fuse observations across this multi-layered representation while explicitly modeling the multi-view consistency of the detections of each object. In order to efficiently search for the target object, we formulate an objective function balancing exploration of unobserved or uncertain regions with exploitation of scene semantic information. We evaluate HELIOS on the OVMM benchmark in the Habitat simulator, a pick and place benchmark in which perception is challenging due to large and complex scenes with comparatively small target objects. HELIOS achieves state-of-the-art results on OVMM. We also demonstrate HELIOS performing language specified pick and place in a real world office environment on a Spot robot. Our method leverages pretrained VLMs to achieve these results in simulation and the real world without any task specific training.

## 1 Introduction

Consider an autonomous robot tasked with bringing a mug from a coffee table to the kitchen counter in a home. If that robot sees a coffee table but cannot currently detect a mug on it, should it go closer to investigate if the mug is actually present? Or should it look in new parts of the home? An autonomous robot should be able to efficiently reason through this question using environment cues. In addition, the robot should be able to successfully perform this task of language-specified pick and place for mobile manipulation using the observations it accumulates during this search process.

Methods for embodied physical intelligence can accumulate information about a novel scene and act on it though observation history with no explicit scene representation (Stone et al., 2023; Physical Intelligence et al., 2025; Team et al., 2025), only 2D maps (Yenamandra et al., 2023b; Melnik et al., 2023) or 3D scene graphs (Rana et al., 2023; Liu et al., 2024; Honerkamp et al., 2024). However, these methods all assume dense associations between language, observation, and action. Very different representations for long horizon spatio-temporal reasoning have been developed in problems for semantic search where language grounding is sparse (Georgakis et al., 2021; Yokoyama et al., 2023b; Chang et al., 2023). In order to perform mobile manipulation which includes semantic search, reasoning over vision, language, and action must occur simultaneously in both long and short horizons. Low success rates on new benchmarks targeting open vocabulary pick and place tasks in novel environments have demonstrated that combining this long and short horizon reasoning is still an open challenge (Liu et al., 2024; Yenamandra et al., 2023b).

Reasoning jointly over short and long spatio-temporal contexts requires very different policy objectives in addition to the differences in scene representations. Prior work in object search explicitly

manages local and global search problems distinctly (Zheng et al., 2023; Schmalstieg et al., 2023; Li et al., 2022). Search policies must figure out when to switch between local and global reasoning by deciding the likelihood of being close to the target object. In addition to exploring unobserved regions, efficient search policies also exploit semantic information about the scene in order to search more likely locations of the target object first (Chaplot et al., 2020b; Ramakrishnan et al., 2022; Ye et al., 2021; Zhang et al., 2023; Georgakis et al., 2021; Yu et al., 2023; Yokoyama et al., 2023b). This exploration-exploitation tradeoff adds additional complexity to the task of performing object search as a component of mobile manipulation. In particular, incorrect object detections have been identified as a major cause of failure for this task (Yenamandra et al., 2023b; Liu et al., 2024; Melnik et al., 2023). While improvements in object detection can help to mitigate these issues there are challenges which occur in robotics data which are much less prevalent in other settings such relevant objects being very far away or only part of the object being visible in the frame. One approach to combat this is to collect more observations to both improve the quality of views of the object and take into account the multi-view consistency of the detections. This aspect of the exploration-exploitation tradeoff has been overlooked in recent works on mobile pick and place tasks (Yenamandra et al., 2023b; Liu et al., 2024; Melnik et al., 2023).

**Contributions.** We present HELIOS, a hierarchical scene representation and search objective for language specified mobile pick and place tasks in novel environments. We create a hierarchical scene representation using layered 2D value and occupancy maps to efficiently navigate and explore, and sparse collections of 3D Gaussians to represent objects of interest (see fig. 1). We then formulate an objective function on our hierarchical scene representation that balances exploring the scene to find regions which might contain the target object with exploiting observed semantic information. We introduce an uncertainty-weighted object score to take into account the multi-view consistency of the detections of an object before interacting with it. We conduct an ablation study to verify that each of these components increases our method's performance. Through our experiments, we show the contribution of uncertainty-based reasoning over our novel visual representation in improving robust perception in mobile manipulation. We evaluate HELIOS on the HomeRobot Open-vocabulary Mobile Manipulation benchmark (Yenamandra et al., 2023b;a) in the Habitat simulator (Savva et al., 2019b), achieving state-of-the-art results. We use HELIOS in semantic navigation as a stop decision, improving overall search success on the Habitat-Matterport 3D (Ramakrishnan et al., 2021) object search benchmark. We also demonstrate HELIOS performing language specified pick and place in a real world office environment on a Spot robot.

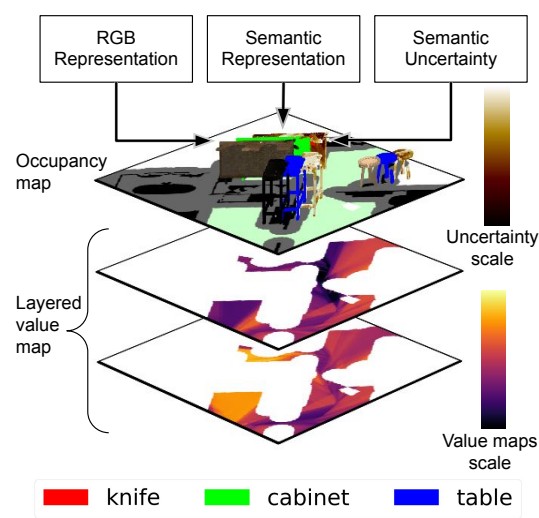

Figure 1: **Hierarchical scene representation.**

## 2 RELATED WORK

**Language-grounded open world pick and place.** Recent advancements in vision and language have opened up challenges in natural language instruction following for robots in novel environments. Many methods focus on parsing complex or ambiguous language and accurately grounding this language to observations made during task execution (Brohan et al., 2023; Rana et al., 2023; Yokoyama et al., 2023a; Shah et al., 2024; Honerkamp et al., 2024). Others focus on improving execution of language specified pick and place skills (Wu et al., 2025; 2024; Shah et al., 2024). However, benchmarks for targeted instantiations of this problem have identified that a major cause of failure in this task is correctly finding and identifying objects for performing pick and place (Yenamandra et al., 2023b; Liu et al., 2024; Melnik et al., 2023). Our work addresses this challenge by modeling the multi-view consistency of object detections, allowing us to only interact with objects once we have obtained enough views that we are confident in the results of the object detection.

**Object search and detection.** To find an object with an RGB camera, that camera needs to record sufficient observations in the environment to correctly identify the object. Active object detection methods obtain additional views of a scene in order to capture an image from which a target object can be correctly identified (Atanasov et al., 2013; Ding et al., 2023; Han et al., 2019). When these observations are accumulated in a map of the environment, it enables a larger scale search problem in which the camera is systematically moved to possible locations in the map. Hierarchical object search methods explicitly perform global and local object search to ensure sensor coverage of the scene (Zheng et al., 2023; Schmalstieg et al., 2023; Li et al., 2022). To perform object search efficiently, semantic information can be used as a prior about where objects are more likely to be (Anderson et al., 2018). This semantic prior naturally yields an exploration and exploitation tradeoff (Chaplot et al., 2020b; Ramakrishnan et al., 2022; Ye et al., 2021; Zhang et al., 2023; Georgakis et al., 2021; Yu et al., 2023; Yokoyama et al., 2023b). In our work, we perform object search and detection as part of pick and place mobile manipulation tasks. Therefore, we construct an objective for switching between global object search and local object detection while simultaneously trading off exploration of the scene and exploitation of semantic information.

**3D Gaussians in robot perception.** 3D Gaussians (Kerbl et al., 2023) have been used in a variety of robotics tasks including SLAM (Matsuki et al., 2024; Keetha et al., 2024), active mapping (Jin et al., 2024; 2025; Jiang et al., 2025), and table-top manipulation (Lu et al., 2024; Zheng et al., 2024). These methods all build a dense 3D representation of the entire scene. Many methods also incorporate open-vocabulary semantic features in 3D Gaussian representations (Zhou et al., 2024; Shi et al., 2024a; Zhang & Lee, 2025; Wilson et al., 2024). In contrast to previous robot perception approaches, we only model target objects of interest with 3D Gaussians, building a sparse 3D map. We adapt Wilson et al. (2024) to perform semantic classification and estimate the associated uncertainty in our sparse 3D Gaussian object map, which forms one layer of our scene representation.

**Language-grounded scene representations.** Language-grounded scene representations can be dense or sparse. Dense open-vocabulary 3D scene representations map vision-language features which can be dynamically queried with language (Rashid et al., 2023; Peng et al., 2023; Kerr et al., 2023; Jatavallabhula et al., 2023; Shi et al., 2024a; Kobayashi et al., 2022). However, these dense 3D representations are not necessarily effective or efficient for performing planning and control. For semantic navigation tasks, dense 2D language-grounded scene representations are more efficient and have been shown to be effective (Huang et al., 2023; Yokoyama et al., 2023b; Georgakis et al., 2021; 2022). For language specified manipulation tasks, instance level information about objects is important (Qian et al., 2024a;b; Zhu et al., 2023; Shi et al., 2024b). To enable mobile manipulation, 3D scene graphs build globally consistent maps of object centric representations needed for manipulation (Gu et al., 2024; Honerkamp et al., 2024; Rosinol et al., 2020; Hughes et al., 2022; Maggio et al., 2024; Chang et al., 2025). Our work builds on this direction in mobile manipulation by using object instance information to construct a sparse map of 3D Gaussians. In our work, we combine this information for manipulation in a hierarchical map with 2D value maps for semantic navigation.

## 3 METHOD

We address the problem of language specified pick and place mobile manipulation tasks in novel environments. To carry out this task, the robot first needs to solve a search problem to find the target object, including correctly identifying the target object. It must then navigate to a suitable grasp position and grasp the object. Finally, it needs to solve another search problem in order to find the place location, and then place the object there in a stable orientation. Note that all of these stages need to be successful, and the robot must also avoid collisions with the environment when navigating and interacting with the objects, so this task is subject to compounding error rates. However the robot can also use information collected in previous stages of the task to aid it later. For example, the search to find the place location can be made more efficient by utilizing information collected when the robot was searching for the target object. In order to collate this information into a useful and efficient format, we propose constructing a hierarchical task-driven map (see Section 3.1) with 2D map layers suitable for the search problems and 3D Gaussians to represent objects in the scene relevant to manipulation. We detail how we explicitly reason over this map to solve a language specified pick and place task in Section 3.2.

### 3.1 Hierarchical Task-driven Map

We construct a hierarchical map with three layers, where each layer corresponds to the three primary tasks that the robot needs to complete. First, to navigate around obstacles to a specified goal location, the robot requires an occupancy map to perform collision free path planning. Second, to efficiently search for objects, the robot can use semantic information in the environment to prioritize exploring unobserved regions which are similar to target locations. Finally, in order to effectively manipulate and perform robust detection of the objects of interest, we model the components of the scene where we expect to perform pick and place with a sparse 3D representation using 3D Gaussians assigned to instances of classes referenced in the instruction.

#### 3.1.1 2D Occupancy Maps

We construct a 2D bird's-eye view (BEV) occupancy map by ground projecting depth measurements. We use this map to perform collision-free path planning to navigate around obstacles to goal locations. We also identify frontiers on the occupancy map, defined as center-points of boundaries between explored and unexplored areas, which will enable us to search unknown map regions.

#### 3.1.2 2D Semantic Value Map

To choose between frontier points, we leverage semantic information about the scene in order to search efficiently by going to areas more likely to contain the target of interest first. We construct a layered semantic value map to enable this frontier-based approach by extending prior work constructing semantic value maps (Yokoyama et al., 2023b) to incorporate multiple search targets. Each layer in our map is a 2D BEV value map constructed by using BLIP-2 (Li et al., 2023) to score the similarity of each observed RGB image to the prompt `Seems like there is a (object) ahead` and fusing the results using a confidence based on the field-of-view cone for each observation. We construct one map layer for the pick location and one for the place location. Since our method is open vocabulary, we can specify the pick location by either referencing the target object directly or by referencing components of the scene where the target object is expected to be.

#### 3.1.3 3D Gaussian representation for modeling objects

In order to enable reasoning about the multi-view consistency of semantic classifications, we represent the objects of interest in the scene using 3D Gaussian Splatting (3DGS) (Kerbl et al., 2023). To increase efficiency over prior applications of 3DGS to robotic tasks (Lu et al., 2024; Zheng et al., 2024), instead of modeling the entire scene with 3D Gaussians we only use them to model parts of the scene which have been detected as objects of interest. We assign Gaussians to object instances, allowing us to reason over objects in the scene instead of individual Gaussians. Our sparse 3DGS representation supports tracking the semantic class probability and semantic class uncertainty for each Gaussian which we use to create a novel uncertainty-weighted object score for each instance.

**Preliminaries – 3D Gaussian representation rendering.** A 3D Gaussian $x(\mu, \Sigma; c, \alpha)$ is defined by its mean position $\mu$, covariance $\Sigma$, color $c$ and opacity $\alpha$, these characteristics can be learned via a rendering loss. A scene is rendered with many of these 3D Gaussians, the final number determined by the task specific conditions in which Gaussians are added and removed. When an image is rendered using 3DGS, the 3D Gaussians comprising the scene representation are first transformed from the world frame to the camera frame and then projected into 2D Gaussians (splats) in the image plane, $x(\mu, \Sigma; c, \alpha) \mapsto \tilde{x}(\tilde{\mu}, \tilde{\Sigma}; c, \alpha)$. Each pixel $i$'s color $Q_i$ is then calculated from the 2D Gaussians using $\alpha$-blending for the $N$ ordered points on the 2D splats that overlap the pixel. For a pixel with position $p_i$ and a 3D Gaussian $x_n$, we first find the opacity $\tilde{\alpha}_n(p_i)$ of the corresponding 2D Gaussian at that pixel position by weighting based on the pixel's distance to the center of the 2D Gaussian with $\tilde{\alpha}_n(p_i) = \alpha_n \cdot k(p_i, \tilde{x}_n)$, where $k(p_i, \tilde{x}_n) = \exp\left(\frac{1}{2}(p_i - \tilde{\mu}_n)\tilde{\Sigma}_n^{-1}(p_i - \tilde{\mu}_n)\right)$. Next, the $N$ Gaussians are ordered based on depth, with $\tilde{x}_1$ being the closest to the camera, and the final contribution for each Gaussian is calculated with $\alpha$-blending to get the final pixel color $Q_i = \sum_{n=1}^{N} c_n \kappa(p_i, \tilde{x}_n; \{\tilde{x}_j\}_{j \in \{1,...,N\}})$ where

$$\kappa(p_i, \tilde{x}_n; \{\tilde{x}_j\}_{j \in \{1,...,N\}}) := \tilde{\alpha}_n(p_i) \prod_{j=1}^{n-1}(1 - \tilde{\alpha}_j(p_i)). \quad (1)$$

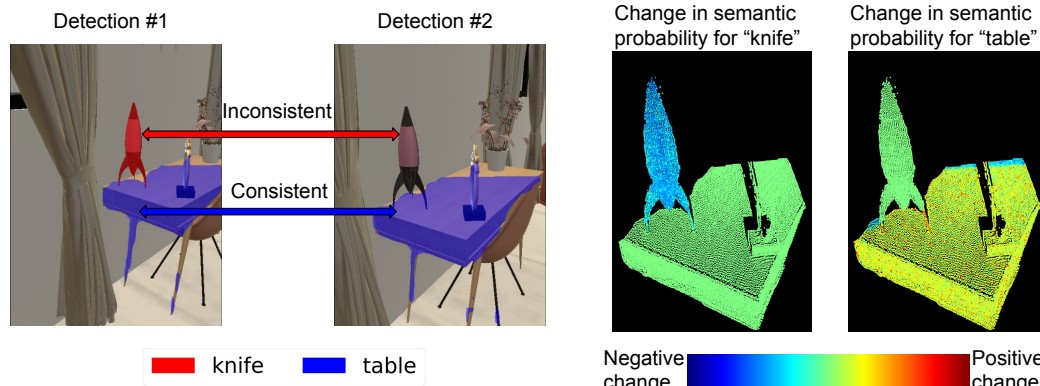

Figure 2: **Example of multi-view fusion.** We show two observations, in the first a toy rocket is incorrectly identified as a knife and the table is correctly identified, in the second the table is again correctly identified. Right of this we show the change in the semantic probability for each class in the 3DGS part of our scene representation when it is updated with the second detection. We can see that the incorrect detection of the object on the table as a knife is not multi-view consistent and so the probability of this object being a knife goes down when we include the second detection. The table is correctly detected across multiple frames so the probability goes up after fusion.

**Preliminaries – Semantic classes for 3D Gaussian representation.** We represent the semantic class scores with our 3DGS model in addition to color. Following Wilson et al. (2024), we explicitly model the distribution of semantic estimates of each Gaussian using the categorical distribution. This distribution is then updated using its conjugate prior, the Dirichlet distribution. Note that this method requires specifying number of object classes at the start of the episode. However, any amount of classes can be specified, so this approach supports open-vocabulary mobile manipulation. The probability density function (PDF) of the Dirichlet distribution is given by

$$f(\theta_n|\boldsymbol{\gamma}_n) = \frac{1}{B(\boldsymbol{\gamma}_n)} \prod_{c=1}^{C} \theta_{n,c}^{\gamma_n^c-1}. \tag{2}$$

where $B$ is the multivariate beta function and $C$ is the number of classes. In our case, $\theta_n$ is the categorical distribution for the Gaussian $x_n$. The concentration parameters, $\boldsymbol{\gamma}_n = (\gamma_n^1, ..., \gamma_n^C)$, of the Dirichlet distribution can be updated after each measurement using Bayesian Kernel Inference as follows (Wilson et al., 2024)

$$\gamma_n^c \leftarrow \gamma_n^c + \sum_{i=1}^{N} y_i^c \kappa(p_i, \tilde{x}_n; \{\tilde{x}_j\}_{j \in \{1,...,N\}}), \tag{3}$$

where $y_i^c$ is 1 if $p_i$ is of class $c$ and 0 otherwise and $\kappa(\cdot)$ is defined in eq. (1).

Then, for a 3D Gaussian $x_n$ and class $c$, the expected probability of $x_n$ being of category $c$ and its variance is given by

$$\mathbb{E}[\theta_n^c] = \frac{\gamma_n^c}{\sum_{j=1}^{C} \gamma_n^j}, \quad \text{Var}[\theta_n^c] = \frac{\mathbb{E}[\theta_n^c](1 - \mathbb{E}[\theta_n^c])}{1 + \sum_{j=1}^{C} \gamma_n^j}. \tag{4}$$

The variance can be considered a measure of the pixel-wise uncertainty of that class score based on the multi-view consistency. During rendering we use $\mathbb{E}[\theta_n^c]$ and $\sqrt{\text{Var}[\theta_n^c]}$ in place of the color parameter for rendering the semantic class scores and uncertainty, respectively. Figure 2 shows an example of how the semantic class score is updated when we obtain a new measurement.

**Preliminaries – Information gain.** Using the Dirichlet distribution to model the semantic state of the Gaussians allows us to find the entropy of the concentration parameters (Lin, 2016)

$$H(\theta_n) = \log B(\boldsymbol{\gamma}_n) + (T(\boldsymbol{\gamma}_n) - C)\psi(T(\boldsymbol{\gamma}_n))$$
$$- \sum_{c=1}^{C} (\gamma_n^c - 1)\psi(\gamma_n^c), \tag{5}$$

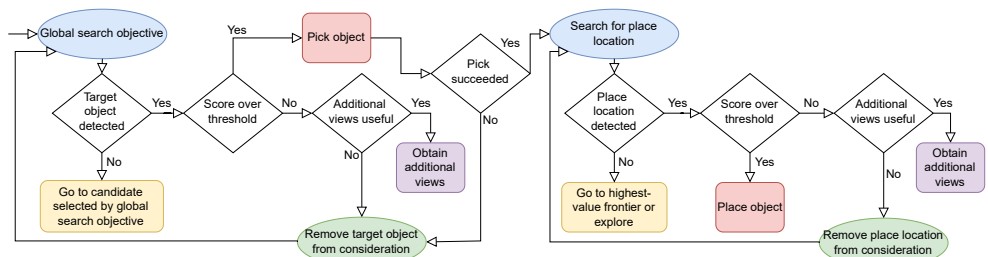

Figure 3: **Method flow chart for HELIOS.**

where $T(\gamma_n) := \sum_{c=1}^{C} \gamma_n^c$ and $\psi$ is the digamma function.

If we obtain a set of new observations, $Y = \{y_1, ..., y_m\}$ at poses $P = \{p_1, .., p_m\}$ then the information gain is

$$\text{IG}(\theta_n, Y|P) = H(\theta_n) - H(\theta_n|P, Y). \tag{6}$$

Given $P$ and $Y$, $H(\theta_n|P, Y)$ can be found by updating $\theta_n$ and then calculating the updated entropy.

**Instances for object-level reasoning.** We assign 3D Gaussians to instances so we can reason about objects. Because the objects are not always perfectly segmented this assignment is done by clustering in 3D within Gaussians which have the same most likely semantic class. To prevent the time requirements becoming intractable for large scenes, we detect which Gaussians are updated for a new observation and only perform the clustering with these Gaussians and any other Gaussians within the same instance.

Using these instances we can reason over the set of objects our representation is modeling, let us call this set $\mathcal{O}$. Each object in $\mathcal{O}$ consists of 3D Gaussians belonging to the same instance, and the class of this object is given by the most common highest-probable class among the 3D Gaussians belonging to that instance, i.e. for $o_i \in \mathcal{O}$, its class is given by $\text{mode}_{\theta \in o_i} \left( \text{argmax}_{c \in \{\text{classes}\}} \mathbb{E}[\theta_n^o] \right)$.

For each object $o_i \in \mathcal{O}$ we also define the class score $S_c := \frac{1}{|o_i|} \sum_{\theta_n \in o_i} \mathbb{E}[\theta_n^c]$, that is, the mean probability of the 3D Gaussians which make up the instance $o_i$ being of class $c$. Likewise, we define the uncertainty $U_c := \frac{1}{|o_i|} \sum_{\theta_n \in o_i} \sqrt{\text{Var}[\theta_n^c]}$.

**Uncertainty-weighted object score** To determine whether we are confident in our estimate of an object's class we define our uncertainty-weighted object score, which takes into account both the class score and uncertainty (balanced by a hyper-parameter $\alpha_{cs}$) for an object $o_i \in \mathcal{O}$ for class $c$:

$$\Psi_c(o_i) := S_c(o_i) - \alpha_{cs} U_c(o_i). \tag{7}$$

That is, the lower bound of the $\alpha_{cs}$-sigma estimate of $o_i$.

### 3.2 HIERARCHICAL SEARCH

We plan over our hierarchical scene representation in a zero-shot manner, searching for the pick location using our global search objective to balance between exploring new frontiers and exploiting semantic information. Once we detect a target object we use our uncertainty-weighted object score to decide whether we are confident enough in the classification to attempt to grasp it. Once the target object has been picked up we perform a similar search procedure until we are confident we have found the place location. Figure 3 shows the logical flow of our method.

**Global search objective.** Our global search objective balances exploring new frontiers with exploiting detections of candidate pick locations. First we introduce some new notion, let $\mathcal{A} \subset \mathcal{O}$ be the set of objects whose class is that of the pick location and let $\mathcal{F}$ be the set of frontiers.

First, we will evaluate the benefit of searching for a detected object. We can work out whether obtaining additional views $Y$ from poses $P$ of candidate pick location $a_i \in \mathcal{A}$ is likely to be informative by considering the information gain (IG). We obtain the proposed poses as described in

the local search section, but we do not have the observations $Y$ unless we move to these poses. In the case of search, we prioritize avoiding false negatives more than false positives since ultimately an effective search policy should provide coverage of the full search space. Thus, we propose an optimistic approach where we assume the best-case scenario that all the observations in $Y$ classify $a_i$ as the pick location $a$. Specifically, we define the estimated information gain as $\text{IG}_a(a_i|P, Y^*) := \sum_{\theta_n \in a_i} H(\theta_n) - H(\theta_n|P, Y^*)$, where $Y^*$ classifies $a_i$ as class $a$. We will drop the condition and just write $\text{IG}_a(a_i)$ for brevity. We can then combine the class score and the IG by multiplying them, i.e. $S_a(a_i)\text{IG}_a(a_i)$, to get a measure of how much we want to search a candidate pick location $a_i$.

This information gain weighted object score allows us to compare candidate objects to each other, but we also want to be able to compare them to frontiers. When we choose a frontier $f_i \in \mathcal{F}$, we store its location and current score from our value map, denote this $F_0(f_i)$. During global planning, the first time each $a_i \in \mathcal{A}$ is detected we store the initial class score, $S_{a0}(a_i)$, the initial information gain, $\text{IG}_{a0}(a_i)$ as well as its initial center position. Then, we want to find the best candidate object while taking into account the distance to the frontier. Explicitly, let $\mathcal{F}'$ be the set of previously chosen frontiers. Then we can calculate an estimated value for a previously chosen frontier $f_i' \in \mathcal{F}'$ based on its proximity to detected candidate objects as:

$$V_0(f_i') := \max_{a_i \in \mathcal{A}} \left( S_{a0}(a_i)\text{IG}_{a0}(a_i) - \alpha_d \text{dist}(a_j, f_i') \right) \tag{8}$$

where $\alpha_d$ is a hyper-parameter which controls the relative importance of candidate object score to distance and $\text{dist}(a_j, f_i')$ is the Euclidean distance between the stored center of $a_j$ and $f_i'$.

Given this association between previous frontiers and candidate object scores we can find an association between frontier scores and candidate object scores by averaging the ratio of this new score to the frontier score over all the previous frontiers:

$$F_0 := \frac{1}{|\mathcal{F}'|} \sum_{f_i' \in \mathcal{F}_p} \frac{V_0(f_i')}{F_0(f_i')} \tag{9}$$

This allows us to associate a frontier $f_i$ with a candidate object score by multiplying its score $F(f_i)$ by $F_0$. We also take into account distance to form the following score function for $r_i \in \mathcal{A} \cup \mathcal{F}$:

$$V(r_i) := \begin{cases} S_a(r_i)\text{IG}_a(r_i) - \alpha_d \text{dist}(r_i) & \text{if } r_i \in \mathcal{A} \\ F(r_i)F_0 - \alpha_d \text{dist}(r_i) & \text{if } r_i \in \mathcal{F} \end{cases} \tag{10}$$

where $F(f_i)$ is the current score from our value map for $f_i \in \mathcal{F}$ and $\text{dist}(r_i)$ is the Euclidean distance from the agent to the center point of $r_i$.

**Local search.** When local search is performed for a candidate pick location, we generate gaze point positions in a contour around the 2D ground-projection of the 3D Gaussians making up our representation of that location. We use our occupancy map to discard gaze points in occupied regions and also remove any gaze points where there is an occupied region over a certain height in the way between the gaze point and the pick location. The orientation of a gaze point is set so that the agent will look towards the center of the object in the ground-plane and the highest point on the object. The robot then goes to each gaze point, starting from the closest. After performing local search for a candidate pick location we mark it as visited and no longer consider it a candidate for local search.

When obtaining additional views for a candidate target object or place location, we generate gaze points in the same way but only go to the one. The one the robot goes to is the one that would result in the highest uncertainty-weighted object score if the object was detected as the class it is believed to be from that viewpoint (i.e. detected as the target object class when considering a target object candidate, or as the place location class when looking for the place location). We choose to only visit one gaze point at a time in this case because at this stage we solely focus on determining the object's class, whereas when searching the pick location the viewpoints need to provide good enough coverage of the location to see if the target object is present and so we do not want to just choose the views which are most informative about the object's class.

# 4 EXPERIMENTAL RESULTS

## 4.1 OPEN VOCABULARY MOBILE PICK AND PLACE IN A NOVEL ENVIRONMENT

**Dataset and benchmark.** We evaluate HELIOS on the validation split of the Home Robot OVMM benchmark (Yenamandra et al., 2023b;a) which uses scenes from the Habitat Synthetic Scenes Dataset (HSSD) (Khanna et al., 2024) in the Habitat simulator (Szot et al., 2021) and consists of 1199 episodes. In this benchmark, the robot must carry out an instruction of the form "Move (object) from the (start_receptacle) to the (goal_receptacle)" in an unknown environment. An oracle pick skill is provided, and we use a simple heuristic place skill to drop the object above the goal_receptacle.

**Metrics.** We report the following metrics from the OVMM benchmark (Yenamandra et al., 2023b;a) indicating the success of each phase of the task: **FindObj** if the robot is ever close enough to the object, **Pick** if the robot successfully picks up the object, **FindRec** if the robot is ever close enough to a goal_receptacle after picking up the object. We additionally report **Place** which indicates if the robot placed the object on the goal_receptacle and the object remained stationary on the goal_receptacle after the set wait period. We also report the success rate (**SR**) as defined in the OVMM benchmark – if all of these stages succeeded without collisions, then episode is considered a success.

**Baselines and ablations.** We evaluate the performance of HELIOS compared to the HomeRobot (Yenamandra et al., 2023b) baseline agents and MoManipVLA (Wu et al., 2025). HomeRobot provides modular implementations of the skills required to carry out the OVMM task, we compare to the results for their reported configurations. Additionally, to isolate the effects of our hierarchical scene representation and global search objective, we include the following ablations of our method:

- **Trusting agent**: this agent uses the same 2D maps and methods for local navigation and place as our full method, but without the 3D portion of our hierarchical scene representation, our gaze points and global search objective. It goes to the frontier with the highest value for the start_receptacle until it detects an object (fully trusting the output of the object detector), at which point it picks up the object. If the pick succeeds, it then goes to the frontier with the highest value for the goal_receptacle until it detects a goal_receptacle, at which point it places the object on it.

- **W/o global search objective**: this agent uses everything from our full method except for the global search objective. Instead, it always prioritizes searching candidate objects over going to frontiers.

- **HELIOS**: our full method, which uses our global search objective to balance when to collect views of a detected start_receptacle and when to go to a frontier.

**Pick Attempts.** In the OVMM benchmark, the agent is allowed an unlimited number of pick attempts. We report results for our method and its ablations with limited numbers of pick attempts (1 and 5) as well as unlimited attempts. With limited pick attempts, if the agent exceeds the limit, we set all metrics for that episode to 0. A benefit of our hierarchical objective is the incorporation of retry logic when we move back and forth between global and local reasoning. In contrast, the baselines do not re-attempt picking. In Table 1, we see that allowing 5 pick attempts provides a significant improvement over 1 pick attempt for HELIOS in all metrics. However, the further benefit of unlimited pick attempts is marginal.

Note that the physical process of grasping the object is not modeled during pick attempts in the OVMM benchmark. The pick action only fails when the target object is not in frame, revealing ground truth information about the scene. Thus, our method has access to ground truth information not accessed by the baselines (which in the real world only corresponds to our method making additional observations) when attempting greater than 1 pick attempt.

**Results.** Table 1 shows the results of our benchmarking and ablation study. Our full method limited to 1 pick outperforms the baselines on all metrics except for FindObj. Adding our hierarchical scene representation and gaze points improves performance compared to our trusting agent, and adding our global search objective results in further improvement for all metrics. This supports our claims that our hierarchical scene representation and global search objective are beneficial for this task.

Table 1: **Ablation study for components of our method**, with comparison to using the HomeRobot (Yenamandra et al., 2023b) baseline agents and recent method MoManipVLA (Wu et al., 2025) on the val split of the OVMM challenge. For HomeRobot the results are included for different configurations of skills for navigation, gaze and place. E.g. R/N/H uses RL for navigation, no skill for gaze and heuristic skill for place.

| | Method | FindObj | Pick | FindRec | Place | SR |
|---|---|---|---|---|---|---|
| | HomeRobot H/N/H | 28.7 | 15.2 | 5.3 | - | 0.4 |
| | HomeRobot H/R/R | **29.4** | 13.2 | 5.8 | - | 0.5 |
| | HomeRobot R/N/H | 21.9 | 11.5 | 6.0 | - | 0.6 |
| | HomeRobot R/R/R | 21.7 | 10.2 | 6.2 | - | 0.4 |
| | MoManipVLA[1] | 23.7 | 12.7 | 7.1 | - | 1.7 |
| 1 pick | Trusting agent | 13.7 ± 1.0 | 12.3 ± 0.9 | 6.8 ± 0.7 | 2.1 ± 0.4 | 1.3 ± 0.3 |
| | W/o global search objective | 16.8 ± 1.1 | 12.0 ± 0.9 | 6.8 ± 0.7 | 2.6 ± 0.5 | 1.7 ± 0.4 |
| | HELIOS | 23.8 ± 1.2 | **17.2 ± 1.1** | **10.0 ± 0.9** | **3.3 ± 0.5** | **2.5 ± 0.5** |
| 5 picks | Trusting agent | 20.4 ± 1.2 | 18.3 ± 1.1 | 10.2 ± 0.9 | 3.2 ± 0.5 | 1.8 ± 0.4 |
| | W/o global search objective | 27.8 ± 1.3 | 21.2 ± 1.2 | 12.8 ± 1.0 | 4.9 ± 0.6 | 2.3 ± 0.4 |
| | HELIOS | **39.2 ± 1.4** | **28.7 ± 1.3** | **17.4 ± 1.1** | **5.8 ± 0.7** | **3.1 ± 0.5** |
| Unlim. | Trusting agent | 21.9 ± 1.2 | 19.3 ± 1.1 | 10.8 ± 0.9 | 3.3 ± 0.5 | 1.8 ± 0.4 |
| | W/o global search objective | 29.6 ± 1.3 | 22.0 ± 1.2 | 13.2 ± 1.0 | 5.0 ± 0.6 | 2.3 ± 0.4 |
| | HELIOS | **42.3 ± 1.4** | **30.5 ± 1.3** | **18.6 ± 1.1** | **6.3 ± 0.7** | **3.2 ± 0.5** |

Table 2: **Adaptation of our method as a stop decision for semantic object goal navigation.** We compare to the original implementation of VLFM, as well as a variant that removes the filtering of detections of objects which are at the sides of images which VLFM uses.

| Method | SPL ↑ | SR ↑ |
|---|---|---|
| VLFM without detection filtering | 29.6 | 50.4 |
| VLFM (Yokoyama et al., 2023b) | **30.4** | 52.5 |
| VLFM with our stop decision | 28.4 | **54.0** |

The place skill is a major cause of failure for our method. We used a simple approach of dropping the `object` above the highest detected point in a region in front of the agent. Because we did not adjust the orientation of the gripper before dropping, we qualitatively observed that the object sometimes rolled off the the `goal_receptacle`. Due to the modularity of HELIOS, we could incorporate other modular solutions to picking without changing our novel contributions.

## 4.2 SEMANTIC OBJECT SEARCH STOP DECISION

**Experiment setting.** We investigate the ability of our sparse 3DGS scene representation and associated uncertainty-weighted object score to contribute to robust object detection in the stopping decision during semantic object search. We replace the stop decision in VLFM (Yokoyama et al., 2023b), a leading modular semantic object search method, with our approach. Specifically, when a potential target object is identified the robot goes to generated gaze points around the object and then only stops if the uncertainty-weighted object score is high enough. We evaluate on the validation split of the HM3D dataset (Ramakrishnan et al., 2021), consisting of 2000 episodes.

**Metrics.** Following VLFM (Yokoyama et al., 2023b) we report the Success Rate (SR) and Success weighted by inverse Path Length (SPL). The SPL is a measure of the robot's efficiency, for a successful episode it is given by the ratio of the length of the shortest successful path for that episode to the path the robot took. It is zero for unsuccessful episodes.

**Results.** Table 2 shows the results. We can see that using our stop decision improves the success rate by 1.5% compared to VLFM and 3.6% compared to VLFM without detection filtering, the proposed

---

[1]We use the reported result for their method without GT semantics for a fair comparison. They do not specify which split of the dataset they use for their evaluation so we assume they use the val split as is standard.

approach in (Yokoyama et al., 2023b) to perform robust object detection. SPL decreases in both cases since our method collects additional views of the objects, prioritizing accuracy over efficiency.

### 4.3 HARDWARE DEMONSTRATIONS

We demonstrate HELIOS on a Boston Dynamics Spot robot in a real-world office environment. In these experiments, we utilize the Spot API to perform grasping and to navigate to the waypoints output by our path planner. We also utilize Bochkovskii et al. (2025) for monocular depth estimation. Videos of these demonstrations are provided in the supplementary material.

## 5 CONCLUSION

We present HELIOS, a hierarchical scene representation and associated search objective, to perform language-specified pick and place mobile manipulation. HELIOS achieves state-of-the-art results on the Open Vocabulary Mobile Manipulation (OVMM) benchmark (Yenamandra et al., 2023a;b) and improves the success rate for modular approaches to semantic object search when used as a stop decision. We demonstrate HELIOS performing language-specified pick and place in a real-world office environment with a Spot robot.

**Limitations.** The performance of HELIOS is limited by errors during execution of subskills including collision avoidance and physical placing which can be improved by integrating better component methods for physical subskills in future work. In addition, since we restrict the total time for executing the pick and place tasks in our work, all of our metrics measure success within a restricted period of time. Therefore, another avenue for increasing performance is by optimizing the choice of gaze points during local search. Filtering for informative gaze points or considering the information gain when generating the gaze points could enable us to achieve improved confidence during local search with fewer total gaze points. Reducing the number of gaze points would allow additional time to enable exploration of more regions in the environment.

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

# 6 APPENDIX

We provide implementation details of our method including our hyper-parameter choices (Section 6.1), compute details (Section 6.2), an additional ablation using ground truth semantics (Section 6.3) and details including licenses of existing assets we use in this work (Section 6.7).

## 6.1 IMPLEMENTATION DETAILS

**Hyperparameters**  The value of the hyper-parameters used in our experiments are given in Table 3.

**Once the target object has been detected**  First we introduce some new notion, let $\mathcal{G} \subset \mathcal{O}$ be the set of objects whose class is that of the target object and $\mathcal{A} \subset \mathcal{O}$ be the set of objects whose class is that of the place location.

Once a candidate target $g_i \in \mathcal{G}$ has been detected we check if it's uncertainty-weighted object score (given by eq. (7)) is over some threshold $\tau_g$, and if $\Psi_g(g_i) \geq \tau_g$ we will treat $g_i$ as the target object.

If $\Psi_g(g_i) < \tau_g$ we can calculate the class score, $S'(g_i)$, and the uncertainty, $U'(g_i)$, if we take $m$ observations $Y$ from poses $P$ and again assume the best-case scenario that each classified $g_i$ as class $g$. Then we can obtain the class score this would give us as

$$\Psi'_g(g_i) := S'(g_i) - \alpha_{cs}U'(g_i). \tag{11}$$

When deciding where to obtain additional views, we consider both if obtaining these views could increase the uncertainty-weighted object score to above the threshold and if the increase in uncertainty-weighted object score is larger than a threshold $\tau_{inc}$. This second condition is so that the agent can obtain views of objects which have not been observed much, and so will have a lower uncertainty-weighted object score due to higher uncertainty. If $\Psi'_g(g_i) \geq \min(\tau_g, \Psi_g(g_i) + \tau_{inc})$ then we will obtain the additional observations of $g_i$, otherwise we return to global search.

After the target object has been grasped, we use the same formulation to decide whether something is the correct class for a place location as we do for deciding whether to grasp a target object but potentially with a different threshold. That is, if we have seen a candidate place location $b_i \in \mathcal{B}$ we first check if $\Psi_b(b_i) \geq \tau_b$ and if so we go there to place the target object, otherwise we check if obtaining additional views satisfies $\Psi'_b(b_i) \geq \min(\tau_b, \Psi_b(b_i) + \tau_{inc})$ and if so we obtain them. If not or if there is no candidate $b_i$ we go to the frontier with the highest value for the place location.

Table 3: **Hyperparameters.** We provide a list of the hyper-parameters of our method with a description and the value used in our experiments. Some hyperparameters are only referenced in the supplementary material and not in the main paper.

| Name | Description | Value |
|------|-------------|-------|
| $\alpha_{cs}$ | Weighting of uncertainty for uncertainty-weighted object score | 1 |
| $\alpha_d$ | Weighting of distance term for global search objective | 0.001 |
| $\tau_g$ | Threshold for uncertainty-weighted object score to pick up an `object` | 0.5 |
| $\tau_b$ | Threshold for uncertainty-weighted object score to place on a `goal_receptacle` | 0.5 |
| $\tau_{inc}$ | Minimum change in uncertainty-weighted object score that would cause us to look at an `object` or `goal_receptacle` | 0.05 |
| $od_a$ | Threshold for object detector confidence for `start_receptacle` class | 0.35 |
| $od_g$ | Threshold for object detector confidence for `object` class | 0.25 |
| $od_b$ | Threshold for object detector confidence for `goal_receptacle` class | 0.45 |
| $cs_a$ | Class score for an object to be considered a candidate `start_receptacle` for the global search objective | 0.3 |
| $cs_g$ | Class score for an object to be considered a candidate `object` for deciding whether to obtain additional views | 0.3 |
| $cs_b$ | Class score for an object to be considered a candidate `goal_receptacle` for deciding whether to obtain additional views | 0.3 |
| $\alpha_{cpa}$ | Absolute concentration parameter update scaling | 3 |

**Path Planner** We modify the fast marching squared (Garrido et al., 2007) motion planner from Home Robot OVMM's baseline (Yenamandra et al., 2023a) to generate navigation actions from the map and the goal pose. Similar to the baseline, our planner also builds the arrival-time map with velocity directly proportional to the distance from the closest obstacle, which balances the efficiency and safety of the motion plan. However, to account for the fine navigation actions required for mobile manipulation, we make 3 modifications to the baseline: 1. Our planner doubles the resolution of the map at 2000 x 2000 cells of 2.5cm x 2.5cm, as the map is directly derived from the depth observations instead of being predicted through a neural network as in the baseline (Chaplot et al., 2020a). 2. Our planner supports continuous actions of moving forward $[0.1m, 1.0m]$ or rotating $[5°, 30°]$, as opposed to fixed actions of moving forward $0.3m$ or rotating $30°$ from the baseline. 3. Our planner explicitly verifies that all intermediate positions for a forward move are collision-free, greatly improving safety around tighter choke-points common in home environments.

**Modifications to 3DGS semantic update** We apply a scaling $\alpha_{cpa}$ directly to the concentration parameter update to control the speed of this update, which corresponds to each observation being repeated $\alpha_{cpa}$ times.

**Additional details of 3DGS instance creation** We spatially cluster gaussians into instances by putting the gaussians in a voxel grid based on the gaussian's center, clustering them by connected components of neighboring voxels, and assigning instance labels to the clusters based on previous assignments. First, we put gaussians of the same semantic label into a grid of 0.5m x 0.5m x 0.5m (adequate due to the spatial sparsity of relevant objects) voxels aligned with the odometry coordinate frame. Then, we take the connected components on the graph of 26-connected voxels containing gaussians. Finally, we assign instance labels to each cluster by taking the minimum of previous instance labels over all gaussians in the cluster. If no gaussian in a cluster previously had an instance label, we assign (maximum instance label over all gaussians) + 1. In practice, this is implemented as a sequence of $\frac{max\_object\_size=10m}{voxel\_size=0.5m} = 20$ min pooling operations on a voxel grid neighborhood graph (Simonovsky & Komodakis, 2017) using the pytorch geometric library (Fey & Lenssen, 2019). Note we perform the above procedure with only the Gaussians which were updated by the last measurement or which were assigned to the same instance as any of these updated Gaussians.

**Gaussian creation** We detect when a new observation represents data which is not already part of our scene representation using the depth error. When an observation is taken, we first make a mask of the pixels which have been detected as an object of interest. Within this mask, we calculate the absolute difference between the measured depth and the rendered depth. We then mask this difference again to keep only the parts where the measured depth is over 0. We find the parts of this difference which are over 1m or over 0.001m and remain after an erosion operation, and create a new Gaussian for each of them. Each Gaussian's position is initialized using the measured depth and camera pose to obtain it's 3D location.

**Re-observing previously detected parts of the scene** As we only model parts of the scene with 3D Gaussians we need to detect when we are re-observing an area which is modeled with 3D Gaussians versus looking towards such an area which is occluded. If we did not do this and only updated the representation when an object is detected then we would not include any negative results (i.e. an object not being detected) and thus we would become over-confident in the classes of objects. One possibility would be to just update if there are any 3D Gaussians in the viewing direction as if they are occluded the new Gaussians should be placed on the occluding object not on the original object, however this is inefficient. Thus we render the depth of our 3D Gaussian scene representation in the viewing direction and then find the pixels in the measured depth image with less than 0.5m of difference to this rendering and finally perform a morphological transformation to close small holes. We then only update the 3D Gaussians using the rendering which lies within this mask.

**Expanded explanation of how we calculate information gain** When updating the global objective score we use

$$\text{IG}_o(o_i|P, Y^*) := \sum_{\theta_n \in o_i} H(\theta_n) - H(\theta_n|P, Y^*). \tag{12}$$

To obtain $Y^*$, for each $\theta_n \in o_i$ we create a copy of the associated 3D Gaussian but with the semantic class probabilities set to 1 for the class $o$ and 0 for all other classes, then render using these

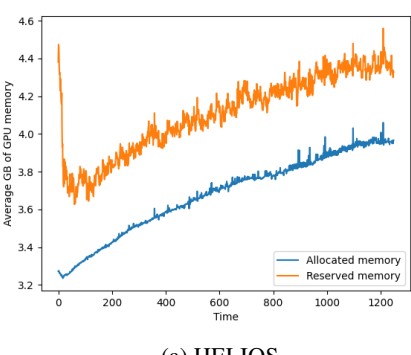

(a) HELIOS

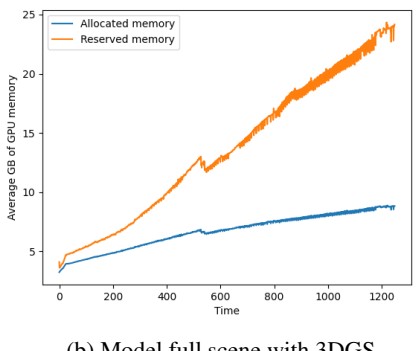

(b) Model full scene with 3DGS

Figure 4: **GPU memory used to model just objects as in HELIOS verses to model the whole scene with the rest of the method unchanged.**

parameters at pose $P$ – this rendered image is used as $Y^*$. Then using $Y^*$ we update a copy of the concentration parameters using Eq. 3 and re-calculate the entropy using the updated concentration parameters with Eq. 5 to obtain $H(\theta_n|P, Y^*)$.

**Object detector**   We use the DETIC (Zhou et al., 2022) object detector as implemented in the HomeRobot codebase. We set separate thresholds for the detections for each class, with the thresholds for the `object` and `start_receptacle` a bit lower than the default used by HomeRobot (0.45) as our method is designed to filter out false positives but does not address false negatives as shown in Table 3.

## 6.2 COMPUTE RESOURCES

The experiments presented in this paper ran in a cluster with 8 nodes, each with a 2080ti GPU with 16GB of VRAM and 32GB of RAM. Our method only runs on one of these GPUs at a time, we use the cluster to parallelize experiments. Each full run of our method or its ablations on the val split took around 288 GPU hours for 1199 episodes.

Figure 4 shows the GPU usage over time averaged across 50 episodes. We show this for both our HELIOS and a version which models the full scene with 3D Gaussians but is otherwise unchanged to show the difference in the memory requirements for these scenarios. Note that the version which models the full scene is still limited to only modeling observations from the robot, it just models all parts of the observations such as the background and objects which are not task-relevant as opposed to modeling only parts of the scene believed to be task-relevant objects. HELIOS is run on a 2080ti GPU as in our main experiments. The version modeling the full scene is run on a L40 with 48GB of VRAM and 64GB of RAM to allow for higher memory requirements. The GPU cache is cleared at the end of each episode using `torch.cuda.empty_cache()`.

We measure the allocated memory just after the scene update to give an indication of the requirements for just keeping the scene representation in GPU memory, and also give the reserved memory to give better indication of the peak memory usage such as during updates or Information Gain calculations. The allocated and reserved memory are both obtained from PyTorch (using `torch.cuda.memory_allocated()` and `torch.cuda.memory_reserved()`, respectively). Thus they represent the memory usage of our entire method not just the 3DGS portion of our scene representation, nevertheless because the rest of the method is unchanged when running these experiments they directly show the result of this change.

As can be seen in Figure 4, modeling just the objects is more efficient than the full scene, especially as time increases. As well as showing the overall trend, we report precise numbers both for the maximum of the curves shown in the graphs and for the maximum reserved memory. The maximum of the average allocated memory is 4.1GB for HELIOS and 8.9GB for the full scene. For the average reserved memory the maximum is 4.6GB for HELIOS and 24.3GB for the full scene. The maxi-

Table 4: **Ablation study for including ground-truth semantics.** We show the performance increase from using ground-truth semantics (with gt) for both our trusting agent, which does not reason about the uncertainty of object detections, and our full method HELIOS, which does. We show the results for our methods with unlimited picks. We also include results of the recent method MoManipVLA (Wu et al., 2025) for additional comparison. The standard error of the mean is indicated.

| Method | FindObj | Pick | FindRec | Place | SR |
|---|---|---|---|---|---|
| MoManipVLA | 23.7 | 12.7 | 7.1 | - | 1.7 |
| MoManipVLA with gt | 66.1 | **62.6** | 53.1 | - | 15.8 |
| Trusting agent | $21.9 \pm 1.2$ | $19.3 \pm 1.1$ | $10.8 \pm 0.9$ | $3.3 \pm 0.5$ | $1.8 \pm 0.4$ |
| Trusting agent with gt | $57.5 \pm 1.4$ | $56.5 \pm 1.4$ | $44.7 \pm 1.4$ | $20.9 \pm 1.2$ | $12.8 \pm 1.0$ |
| HELIOS | $42.3 \pm 1.4$ | $30.5 \pm 1.3$ | $18.6 \pm 1.1$ | $6.3 \pm 0.7$ | $3.2 \pm 0.5$ |
| HELIOS with gt | $\mathbf{66.3 \pm 1.4}$ | $58.3 \pm 1.4$ | $\mathbf{53.4 \pm 1.4}$ | $\mathbf{29.8 \pm 1.3}$ | $\mathbf{21.0 \pm 1.2}$ |

mum reserved memory found using `torch.cuda.max_memory_reserved()` for is 7.1GB for HELIOS and 43.0GB for the full scene.

## 6.3 ABLATION USING GROUND TRUTH SEMANTICS

We perform an ablation study to show the effect of using ground-truth semantics on performance, the results are shown in Table 4. We can see that our full method our-performs our trusting agent when both use ground truth semantics, this may be due to fact that HELIOS performs local search of detected pick locations whereas our trusting agent doesn't. The gap between the pick success of our trusting agent and our full method is much smaller with ground truth semantics (11.2% without ground truth semantics and 1.8% with ground truth semantics). Likewise, the gap in pick success with and without semantics is much higher for both MoManipVLA and our trusting agent than for HELIOS (49.9% for MoManipVLA, 37.2% for our trusting agent and 27.8% for HELIOS). These results indicate that our full method is less of an improvement when ground truth semantics are used. This makes sense because alleviating issues from imperfect object detections is the main focus of the components of HELIOS which are included in the full method but not in our trusting agent. Addressing this challenge is not necessary when ground truth semantics are provided.

The relatively low overall success rates with ground truth semantics for both MoManipVLA and our method indicate there is still more work required to increase search efficiency and the success rate of physical subskills such as collision-free navigation and place. However the large gap between the results with and without ground truth semantics for MoManipVLA and our trusting agent, especially for the pick skill, still shows that robust object detection is a key bottleneck for this task. While HELIOS still has a performance gap when not using ground truth semantics it takes a step towards addressing this issue.

## 6.4 MORE DETAILED FAILURE ANALYSIS

Figure 5 shows the failure cases breakdown for HELIOS and Figure 6 shows the failure cases when using ground truth semantics. As can be seen, the collisions between the robot and scene are a major cause of failure. This could be addressed by a better local path planner, however we expect that what works well in this simulation environment may not work well in the real world or even in other simulators. Thus we consider improving this aspect of our method to be secondary to some other sources of failure, even if they are less significant in this setting. Failure to find the target object is the second largest cause of failure, and the largest cause of failure without ground truth semantics. Because it is still a large cause of failure even with ground truth semantics this failure seems to be mostly due to inefficient search rather than incorrect object detections. Improving the efficiency of our local search could help with this, for example by optimizing the choice of gaze points as discussed in Section 5. It may also be possible to incorporate future improvements in semantic object navigation to improve the overall efficiency of searching for objects. Finally we see that the place skill is a major failure cause, in particular objects not being placed correctly onto

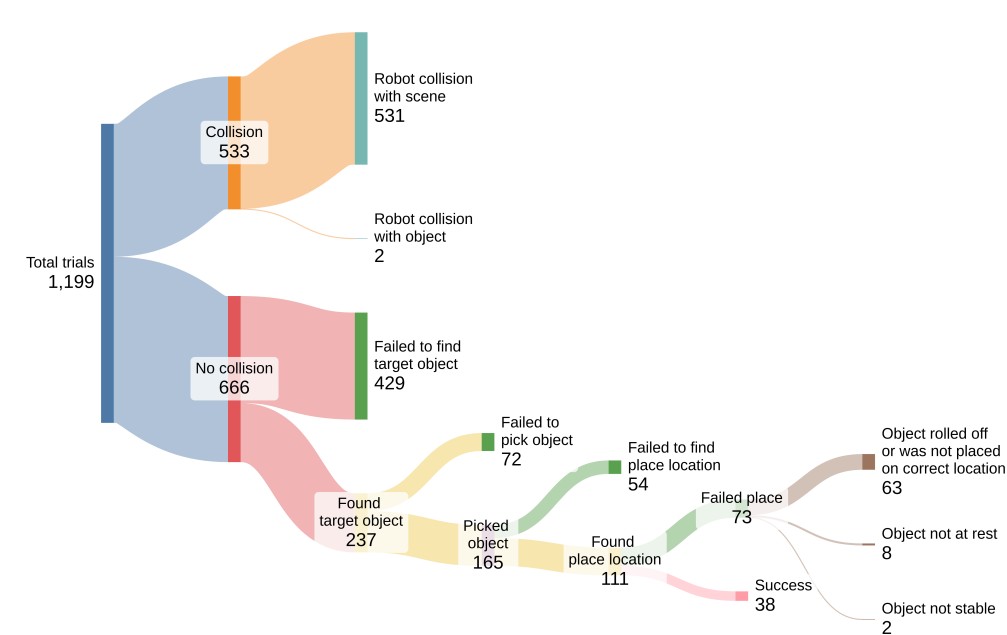

Figure 5: **Failure mode break-down in simulation for HELIOS.**

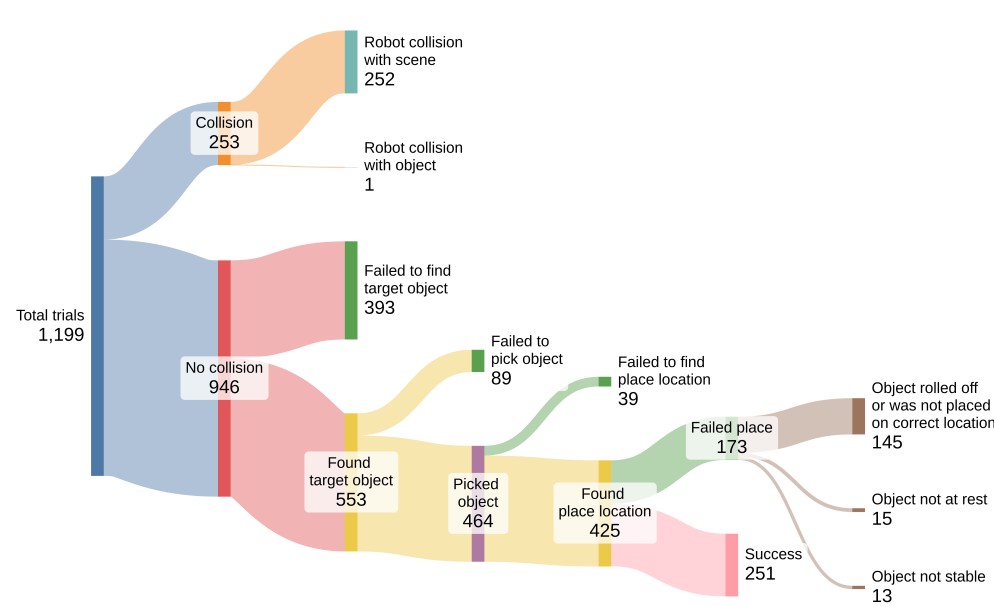

Figure 6: **Failure mode break-down in simulation for HELIOS with ground truth semantics.**

the place location or rolling off (our metrics cannot distinguish between these cases). Note that this remains a major cause of failure when using ground truth semantics.

The difference in the collision rates between HELIOS with and without ground truth could be due to the robot taking longer to find the relevant objects when ground truth is not provided, thus giving more opportunities for it to collide with the environment.

Table 5: **Ablation study for the information gain update assumption on the val split of the OVMM challenge.**

| N picks | Method | FindObj | Pick | FindRec | Place | SR |
|---|---|---|---|---|---|---|
| 1 | 50-50 | $14.5 \pm 1.0$ | $10.5 \pm 0.9$ | $6.6 \pm 0.7$ | $1.9 \pm 0.4$ | $1.5 \pm 0.4$ |
| | Optimistic | $\mathbf{23.8 \pm 1.2}$ | $\mathbf{17.2 \pm 1.1}$ | $\mathbf{10.0 \pm 0.9}$ | $\mathbf{3.3 \pm 0.5}$ | $\mathbf{2.5 \pm 0.5}$ |
| 5 | 50-50 | $23.1 \pm 1.2$ | $17.9 \pm 1.1$ | $12.0 \pm 0.9$ | $3.9 \pm 0.6$ | $2.2 \pm 0.4$ |
| | Optimistic | $\mathbf{39.2 \pm 1.4}$ | $\mathbf{28.7 \pm 1.3}$ | $\mathbf{17.4 \pm 1.1}$ | $\mathbf{5.8 \pm 0.7}$ | $\mathbf{3.1 \pm 0.5}$ |
| Unlim. | 50-50 | $26.0 \pm 1.3$ | $19.5 \pm 1.1$ | $12.7 \pm 1.0$ | $3.9 \pm 0.6$ | $2.2 \pm 0.4$ |
| | Optimistic | $\mathbf{42.3 \pm 1.4}$ | $\mathbf{30.5 \pm 1.3}$ | $\mathbf{18.6 \pm 1.1}$ | $\mathbf{6.3 \pm 0.7}$ | $\mathbf{3.2 \pm 0.5}$ |

## 6.5 Ablation Study on Information Gain Assumption

Table 5 shows an ablation of what assumption about the measurement we use to calculate the information gain. HELIOS uses the Optimistic update which assumes that the measurement will be whatever object we are looking for (so if we think something might be a pick location we assume the measurement will be the class of the pick location). We compare to using a more conservative estimate (50-50) which assigns 50% probability of the measurement being whatever object we are looking for and 50% it being the other/background class. As we can see the optimistic assumption performs much better.

## 6.6 Sensitivity Analysis

Table 6, Table 7 and Table 8 show the effect of some hyper-parameter changes in the 1-pick, 5-picks and unlimited picks cases respectively. Only one hyper-parameter is modified at a time, all unmodified hyper-parameters use the values given in Table 3. These experiments are all performed for our method HELIOS without ground truth semantics.

Table 6: **Sensitivity analysis results for 1 pick.**

| Param | Value | FindObj | Pick | FindRec | Place | SR |
|---|---|---|---|---|---|---|
| $\alpha_{cs}$ | 0.1 | $22.8 \pm 1.2$ | $15.3 \pm 1.0$ | $8.8 \pm 0.8$ | $2.6 \pm 0.5$ | $2.3 \pm 0.4$ |
| | 1 | $23.8 \pm 1.2$ | $\mathbf{17.2 \pm 1.1}$ | $10.0 \pm 0.9$ | $3.3 \pm 0.5$ | $\mathbf{2.5 \pm 0.5}$ |
| | 2 | $\mathbf{24.1 \pm 1.2}$ | $16.6 \pm 1.1$ | $\mathbf{10.6 \pm 0.9}$ | $\mathbf{3.4 \pm 0.5}$ | $2.2 \pm 0.4$ |
| $\alpha_d$ | 0.0001 | $22.8 \pm 1.2$ | $15.4 \pm 1.0$ | $9.0 \pm 0.8$ | $2.9 \pm 0.5$ | $1.8 \pm 0.4$ |
| | 0.001 | $\mathbf{23.8 \pm 1.2}$ | $\mathbf{17.2 \pm 1.1}$ | $\mathbf{10.0 \pm 0.9}$ | $\mathbf{3.3 \pm 0.5}$ | $\mathbf{2.5 \pm 0.5}$ |
| | 0.01 | $23.4 \pm 1.2$ | $16.5 \pm 1.1$ | $9.8 \pm 0.9$ | $2.9 \pm 0.5$ | $2.2 \pm 0.4$ |
| $\tau_g$ | 0.3 | $20.0 \pm 1.2$ | $14.5 \pm 1.0$ | $7.8 \pm 0.8$ | $2.1 \pm 0.4$ | $1.1 \pm 0.3$ |
| | 0.5 | $23.8 \pm 1.2$ | $\mathbf{17.2 \pm 1.1}$ | $10.0 \pm 0.9$ | $\mathbf{3.3 \pm 0.5}$ | $\mathbf{2.5 \pm 0.5}$ |
| | 0.7 | $\mathbf{25.0 \pm 1.3}$ | $16.5 \pm 1.1$ | $\mathbf{10.2 \pm 0.9}$ | $3.0 \pm 0.5$ | $1.8 \pm 0.4$ |
| $\tau_{inc}$ | 0.05 | $\mathbf{23.8 \pm 1.2}$ | $\mathbf{17.2 \pm 1.1}$ | $\mathbf{10.0 \pm 0.9}$ | $\mathbf{3.3 \pm 0.5}$ | $\mathbf{2.5 \pm 0.5}$ |
| | 0.1 | $21.4 \pm 1.2$ | $14.9 \pm 1.0$ | $7.8 \pm 0.8$ | $1.8 \pm 0.4$ | $1.0 \pm 0.3$ |

Table 7: **Sensitivity analysis results for 5 picks.**

| Param | Value | FindObj | Pick | FindRec | Place | SR |
|---|---|---|---|---|---|---|
| | 0.1 | $38.5 \pm 1.4$ | $27.6 \pm 1.3$ | $16.2 \pm 1.1$ | $5.8 \pm 0.7$ | $\textbf{3.3} \pm \textbf{0.5}$ |
| $\alpha_{cs}$ | 1 | $\textbf{39.2} \pm \textbf{1.4}$ | $\textbf{28.7} \pm \textbf{1.3}$ | $\textbf{17.4} \pm \textbf{1.1}$ | $5.8 \pm 0.7$ | $3.1 \pm 0.5$ |
| | 2 | $38.5 \pm 1.4$ | $26.8 \pm 1.3$ | $\textbf{17.4} \pm \textbf{1.1}$ | $\textbf{6.4} \pm \textbf{0.7}$ | $\textbf{3.3} \pm \textbf{0.5}$ |
| | 0.0001 | $36.2 \pm 1.4$ | $26.2 \pm 1.3$ | $15.6 \pm 1.0$ | $5.0 \pm 0.6$ | $2.4 \pm 0.4$ |
| $\alpha_d$ | 0.001 | $\textbf{39.2} \pm \textbf{1.4}$ | $\textbf{28.7} \pm \textbf{1.3}$ | $\textbf{17.4} \pm \textbf{1.1}$ | $\textbf{5.8} \pm \textbf{0.7}$ | $\textbf{3.1} \pm \textbf{0.5}$ |
| | 0.01 | $38.5 \pm 1.4$ | $28.3 \pm 1.3$ | $17.3 \pm 1.1$ | $\textbf{5.8} \pm \textbf{0.7}$ | $3.0 \pm 0.5$ |
| | 0.3 | $34.7 \pm 1.4$ | $26.2 \pm 1.3$ | $14.8 \pm 1.0$ | $4.7 \pm 0.6$ | $1.8 \pm 0.4$ |
| $\tau_g$ | 0.5 | $\textbf{39.2} \pm \textbf{1.4}$ | $\textbf{28.7} \pm \textbf{1.3}$ | $\textbf{17.4} \pm \textbf{1.1}$ | $\textbf{5.8} \pm \textbf{0.7}$ | $\textbf{3.1} \pm \textbf{0.5}$ |
| | 0.7 | $36.6 \pm 1.4$ | $25.0 \pm 1.3$ | $15.9 \pm 1.1$ | $5.2 \pm 0.6$ | $2.5 \pm 0.5$ |
| | 0.05 | $\textbf{39.2} \pm \textbf{1.4}$ | $\textbf{28.7} \pm \textbf{1.3}$ | $\textbf{17.4} \pm \textbf{1.1}$ | $\textbf{5.8} \pm \textbf{0.7}$ | $\textbf{3.1} \pm \textbf{0.5}$ |
| $\tau_{inc}$ | 0.1 | $37.1 \pm 1.4$ | $26.9 \pm 1.3$ | $14.7 \pm 1.0$ | $3.7 \pm 0.5$ | $1.3 \pm 0.3$ |

Table 8: **Sensitivity analysis results for unlimited picks.**

| Param | Value | FindObj | Pick | FindRec | Place | SR |
|---|---|---|---|---|---|---|
| | 0.1 | $42.0 \pm 1.4$ | $29.8 \pm 1.3$ | $17.1 \pm 1.1$ | $6.1 \pm 0.7$ | $3.3 \pm 0.5$ |
| $\alpha_{cs}$ | 1 | $\textbf{42.3} \pm \textbf{1.4}$ | $\textbf{30.5} \pm \textbf{1.3}$ | $\textbf{18.6} \pm \textbf{1.1}$ | $6.3 \pm 0.7$ | $3.2 \pm 0.5$ |
| | 2 | $39.9 \pm 1.4$ | $27.5 \pm 1.3$ | $17.9 \pm 1.1$ | $\textbf{6.8} \pm \textbf{0.7}$ | $\textbf{3.4} \pm \textbf{0.5}$ |
| | 0.0001 | $38.5 \pm 1.4$ | $27.6 \pm 1.3$ | $16.3 \pm 1.1$ | $5.3 \pm 0.6$ | $2.4 \pm 0.4$ |
| $\alpha_d$ | 0.001 | $\textbf{42.3} \pm \textbf{1.4}$ | $\textbf{30.5} \pm \textbf{1.3}$ | $\textbf{18.6} \pm \textbf{1.1}$ | $\textbf{6.3} \pm \textbf{0.7}$ | $\textbf{3.2} \pm \textbf{0.5}$ |
| | 0.01 | $41.7 \pm 1.4$ | $30.4 \pm 1.3$ | $\textbf{18.6} \pm \textbf{1.1}$ | $6.2 \pm 0.7$ | $3.0 \pm 0.5$ |
| | 0.3 | $38.4 \pm 1.4$ | $28.6 \pm 1.3$ | $16.2 \pm 1.1$ | $4.9 \pm 0.6$ | $1.8 \pm 0.4$ |
| $\tau_g$ | 0.5 | $\textbf{42.3} \pm \textbf{1.4}$ | $\textbf{30.5} \pm \textbf{1.3}$ | $\textbf{18.6} \pm \textbf{1.1}$ | $\textbf{6.3} \pm \textbf{0.7}$ | $\textbf{3.2} \pm \textbf{0.5}$ |
| | 0.7 | $38.1 \pm 1.4$ | $25.7 \pm 1.3$ | $16.3 \pm 1.1$ | $5.4 \pm 0.7$ | $2.6 \pm 0.5$ |
| | 0.05 | $\textbf{42.3} \pm \textbf{1.4}$ | $\textbf{30.5} \pm \textbf{1.3}$ | $\textbf{18.6} \pm \textbf{1.1}$ | $\textbf{6.3} \pm \textbf{0.7}$ | $\textbf{3.2} \pm \textbf{0.5}$ |
| $\tau_{inc}$ | 0.1 | $39.8 \pm 1.4$ | $28.2 \pm 1.3$ | $15.5 \pm 1.0$ | $3.8 \pm 0.5$ | $1.3 \pm 0.3$ |

## 6.7 DETAILS OF EXISTING ASSETS USED

Directly-used assets:

- Home Robot OVMM benchmark and code (Yenamandra et al., 2023b;a): MIT License, commit `ede6a67a` (main branch as of submission). `https://github.com/facebookresearch/home-robot`

- Habitat Synthetic Scenes Dataset (HSSD) (Khanna et al., 2024): cc-by-nc-4.0, obtained using Home Robot's download script `https://huggingface.co/datasets/hssd/hssd-hab`

- Habitat (Savva et al., 2019a; Szot et al., 2021; Puig et al., 2023): MIT License, for habitat-lab we used HomeRobot's modified code, for habitat-sim we use `v0.2.5`.

  `https://github.com/facebookresearch/habitat-lab`

  `https://github.com/facebookresearch/habitat-sim`

- VLFM (Yokoyama et al., 2023b): MIT License

  `https://github.com/bdaiinstitute/vlfm`

- gsplat (Ye et al., 2025): Apache License 2.0

  `https://github.com/nerfstudio-project/gsplat`

- SplaTAM (Keetha et al., 2024): BSD 3-Clause License, some code used with modifications rather than directly importing

  `https://github.com/spla-tam/SplaTAM/`

Key assets used in above works that we also use:

- BLIP2 (Li et al., 2023): BSD 3-Clause License, v1.0.2
  https://github.com/salesforce/LAVIS
- DETIC (Zhou et al., 2022): Apache License 2.0, installed via HomeRobot
  https://github.com/facebookresearch/Detic

