# OpenReview forum: "HELIOS: Hierarchical Exploration for Language-grounded Interaction in Open Scenes"
_ICLR.cc/2026/Conference — Submitted to ICLR 2026_

### Official Review · Reviewer_UXNY · 2025-11-01

**Soundness:** 4
**Presentation:** 3
**Contribution:** 3
**Rating:** 6
**Confidence:** 3

**Summary:**

This paper presents HELIOS, a novel method for performing language-specified mobile pick-and-place tasks in open-world environments. HELIOS introduces a hierarchical scene representation combining 2D semantic and occupancy maps with sparse 3D Gaussian object modeling. The method also formulates a global search objective that balances exploration and exploitation via information gain and semantic confidence. The approach is evaluated on the Open-Vocabulary Mobile Manipulation (OVMM) benchmark in the Habitat simulator and demonstrates state-of-the-art performance. Additionally, the method is shown to transfer to the real world zero-shot on a Boston Dynamics Spot robot.

**Strengths:**

- Hierarchical representation: Combines 2D semantic maps and 3D Gaussian instance modeling, enabling robust semantic search and manipulation.

- Well-formulated global search objective: Uses uncertainty-aware object scoring and information gain estimation to balance exploration vs. exploitation.

- Comprehensive experiments: Includes comparisons against strong baselines (HomeRobot, MoManipVLA), ablation studies, limited vs. unlimited pick attempts, and real-world robot demonstrations.

- Zero-shot transfer experiment: Modular design and zero-shot transferability make it a promising foundation for future real-world mobile manipulation systems.

SOTA performance: HELIOS consistently outperforms baselines across all metrics, especially in success rate and robustness to failures.

**Weaknesses:**

- Technical novelty: The primary contributions lie in system integration. Key components—BLIP-2, DETIC, 3D Gaussian Splatting, and frontier-based exploration—are all existing techniques. The global search objective is largely heuristic and lacks rigorous derivation or theoretical insight. The engineering contribution is appreciable; however, technical novelty can be improved.

- The theory of the search strategy:
The proposed global search objective assumes optimistic information gain estimates. There is no theoretical justification or analysis of its performance bounds or optimality. From instinct, the method is reasonable, but a more strict analysis will help to improve the paper.

- Generalization analysis can be more detailed
While the method is claimed to be zero-shot, there is no systematic evaluation of generalization across scene types, instruction types, or camera viewpoints. The robustness to domain shift or novel object categories is unclear.
- Inconsistent or unfair comparisons
The baselines (e.g., MoManipVLA, HomeRobot) may not be using the same level of semantic fusion or multi-view detection. Comparisons may thus favor HELIOS due to richer intermediate representations. More new or related baselines?

**Questions:**

See weakness

---

> ### Author Response · Authors · 2025-11-21
> **W2, W3, W4 (W1 in top-level comment)**
>
> Thank you for your detailed review. Please see our top-level comment for a detailed response to **W1**, we also clarify what we mean by “zero-shot” which relates to **W3**.
>
> **W2**. Providing a proof of the theoretical performance bounds or optimality of our information gain assumption would constitute a new contribution for our work. It is also unclear how to do so with our current formulation without making other simplifying assumptions because we do not know the precise relationship between an object being of a certain class and being detected as that class. To provide some more justification of this choice we now include a small ablation study where we change the assumption when calculating the information gain to give 50% chance of being the object of interest and a 50% chance of being background (please see Section 6.5 in the updated paper). This new assumption performs worse than our current assumption when calculating the information gain.
>
> **W3**. We rely on the evaluation on the OVMM benchmark [1] for any claims of generalizability. This benchmark includes objects from 129 categories, please see their report for details on the object classes and the distribution of classes in particular section D1 in their appendix. Note that in both simulation and the real world we have some control over the viewing angle of the robot, we can change the pitch and roll not just the yaw, we use this to look at the relevant object during local search, however it is true we did not conduct a systematic evaluation of the effect of this aspect on the performance of our method.
>
> **W4**. The idea of applying semantic fusion and multi-view detection to this problem and the implementation of how to do so are a core part of our contribution. We use the same object detector as the HomeRobot baselines and so do not receive an unfair advantage because of advances in open-vocabulary object detection. Please also note that the HomeRobot baselines are fairly recent (2023) [1] and MoManipVLA is from this year [2].
>
>
> [1] Yenamandra, S., Ramachandran, A., Yadav, K., Wang, A., Khanna, M., Gervet, T., ... & Paxton, C. (2023). Homerobot: Open-vocabulary mobile manipulation. NeurIPS
>
> [2] Wu, Z., Zhou, Y., Xu, X., Wang, Z., & Yan, H. (2025). Momanipvla: Transferring vision-language-action models for general mobile manipulation. In Proceedings of the Computer Vision and Pattern Recognition Conference (pp. 1714-1723).

---

### Official Review · Reviewer_wuZW · 2025-11-02

**Soundness:** 2
**Presentation:** 2
**Contribution:** 1
**Rating:** 2
**Confidence:** 2

**Summary:**

The paper proposes HELIOS, a hierarchical exploration and interaction framework for language-specified mobile pick-and-place. HELIOS builds a 3 layer scene representation consisting of (i) a 2D occupancy map, (ii) layered 2D semantic value maps to guide frontier exploration and (iii) a sparse 3D Gaussian object map. Planning is done by a global search objective that trades off exploration and exploitation via an optimistic information-gain estimate. Results are presented on the OVMM and ObjectNav benchmark in Habitat.

**Strengths:**

* The method achieves the state of the art results on the OVMM benchmark.
* The authors also show the transfer of their method to the real world.

**Weaknesses:**

1) The contributions of the paper are very unclear to me. Section 3.1.1 and 3.1.2 are taken from or heavily based on prior work. Most of Sec 3.1.3 is Preliminaries. From what I understand, the main contributions are the integration of the 2D and 3D scene representations, the Uncertainty-weighted object score and the global search objective. If I have gotten it correctly, I believe in its current form, the work falls below the ICLR acceptance bar.

2) While 3DGS is sparse and instance-clustered, it's still very memory and compute intensive based on the details in section 6.2 in the appendix.

3) The Spot demo is qualitative; no success rates, timing, or object set diversity are reported.

4) I would not consider the results in section 4.2 to be the strength of the proposed method since it leads to 1.5% gain in Success but 2.0% drop in SPL compared to the baseline.

**Questions:**

Minor suggestion: The paper’s readability could be improved if there were fewer images taking half width of the page.

---

> ### Author Response · Authors · 2025-11-21
> **W2, W4, Q1 (W1 and W3 in top-level comment)**
>
> Thank you for your detailed review. For **W1** we would argue that our hierarchical scene representation involves more than just “integration of the 2D and 3D scene representations”, for more details please see our top-level comment. **W3** is also addressed in the top-level comment.
>
> **W2**. We think that there may have been some confusion on the computational requirements due to our prior wording in Section 6.2. It is 288 GPU hours total with each episode running on a single GPU not on 8 GPUs. We have updated the wording of this section to make this point clear. If you already understood this point or still have this concern please let us know. We also now provide some additional information about the GPU requirements in Figure 4, showing the efficiency of our representation which only models objects as opposed to the whole scene with 3DGS following feedback from reviewer Yd22.
>
> **W4**. This is still a useful finding because it can allow some control of the trade-off between accuracy and efficiency for already-existing methods (rather than needing to develop a new method that has a different trade-off). We also note that VLFM does some filtering to improve object detection already and if we compare our method to VLFM without this filtering we improve success by 3.6% at a cost of increased path length which only reduces SPL by 1.2%. Thus for semantic navigation methods which do not already include a way to improve the object detections our method may be more useful.
>
> **Q1**. Thank you for your suggestion, we have updated the paper to use full-width figures where possible (figures 2 and 3 have been updated).

---

### Official Review · Reviewer_JyoA · 2025-11-02

**Soundness:** 3
**Presentation:** 3
**Contribution:** 3
**Rating:** 6
**Confidence:** 3

**Summary:**

This paper presents HELIOS, a hierarchical exploration and scene representation framework for language-grounded mobile manipulation tasks in partially observed and open-world environments. The core challenge addressed is performing language-specified pick-and-place tasks when both the spatial environment and semantic cues are only partially observable. HELIOS combines: 2D layered maps, 3D Gaussian splatting, a hierarchical search objective, and an uncertainty-weighted object score into one framework. The system is evaluated on the OVMM benchmark and HM3D semantic navigation tasks, achieving state-of-the-art performance in open-vocabulary mobile manipulation and improved success rates when integrated as a stopping criterion for semantic object search.

**Strengths:**

- The hierarchical integration of 2D and sparse 3D Gaussian maps for exploration and manipulation is novel and elegantly addresses multi-scale reasoning.  And the uncertainty-weighted object score provides a principled way to leverage semantic confidence and multi-view consistency in decision-making.

- The paper is technically solid and mathematically well-founded. The 3D Gaussian semantics formulation with Dirichlet updates, entropy-based information gain, and uncertainty weighting is rigorous.

- Ablation studies (e.g., removing global search, using ground-truth semantics) are extensive and well-designed to isolate each contribution.  Clear experimental validation shows consistent improvement over strong baselines and across different pick-attempt limits.

- This paper aims at addressing a pressing open problem in embodied AI: bridging language grounding, exploration, and manipulation under partial observability.  The zero-shot transfer to real hardware adds strong practical significance.

**Weaknesses:**

- HELIOS mainly combines existing building blocks: frontier-based exploration, open-vocabulary object detection (BLIP-2, DETIC), and 3D Gaussian splatting with Bayesian updates. The integration is well-engineered but does not introduce fundamentally new algorithms.

- The system involves many components (3DGS updates, frontier mapping, uncertainty weighting, search objectives, multi-threshold heuristics).  Although ablations exist, the paper lacks deeper sensitivity analyses on hyperparameters like $\alpha_{cs}$, $\tau_g$, $\tau_{inc}$, and $\alpha_d$, which control trade-offs between exploration and confidence.

- Improvements in success rate (~2–3%) over MoManipVLA, while consistent, are relatively small considering the complexity. Performance is bottlenecked by upstream perception (object detection) and downstream grasping failures - limitations acknowledged but not experimentally analyzed.

- The system is heavy: ~288 GPU-hours for 1199 episodes on 8 GPUs.  Real-time feasibility for on-board deployment is unclear, particularly for dense environments or more complex instructions.

**Questions:**

I am not an expert in this domain, thus I have the following questions. If misunderstanding, please let me know.

Q1: Could you elaborate on how information gain influences exploration behavior in practice? How sensitive is performance to its weighting relative to distance ($\alpha_d$)?

Q2: Does the uncertainty-weighted score ever mislead the agent into revisiting false positives with low variance? Could you visualize failure cases or confidence evolution over time?

Q3: Have you considered learned policies (e.g., RL or imitation learning) to modulate the exploration–exploitation balance instead of fixed weights?

---

> ### Author Response · Authors · 2025-11-21
> **W2, W3, W4, Q1, Q2 (W1 in top-level comment)**
>
> Thank you for your detailed review. Please see our top-level comment for our response to **W1**.
>
> **W2**. We now include a sensitivity analysis in the Appendix (Section 6.6.). We are still running some of the experiments and will update the table in our paper and comment letting you know we have done so when they complete.
>
> **W3**. We do provide some analysis of the object detection bottleneck by conducting an ablation study with ground truth semantics in the Appendix (Section 6.3. Table 4). We agree that the increase in overall success rate is not very large, in part because this is a challenging benchmark (as shown by the overall low success numbers for all methods) which requires competency at multiple distinct sub-skills not all of which we seek to improve with our method (e.g. placing). We hope that our ablation study can still show the benefit of our method design. Also, our modular approach means that we could incorporate future work which addresses these subskills without needing to change the core of our method.
>
> **W4**. Looking at our Section 6.2 we think it was not written in a clear way and thus you may have misunderstood what we meant. It is 288 GPU hours total with each episode running on a single GPU. We reported the total GPUs we used due to standards around transparency of the compute resources required to reproduce work but this was poorly worded by us. We have updated the wording of this section to make this point clear. If you did understand what we meant or still have this issue please let us know. We also now provide some additional information about the GPU requirements in Figure 4, showing the efficiency of our representation which only models objects as opposed to the whole scene with 3DGS following feedback from reviewer Yd22.
>
> **Q1**. We performed an ablation study which investigates changing the assumption we use when calculating the information gain following feedback from reviewer UXNY (Section 6.5 in the Appendix). Changing this assumption had a larger impact on the performance than increasing $\alpha_d$ by an order of magnitude as shown in our sensitivity analysis (Section 6.6).
>
> **Q2**. The uncertainty-weighted object score is only used for the target object and place location, not when searching the pick location for the target object. Thus if an object has a high class score and a low variance the robot will attempt to pick it (if it’s a target object) or place onto it (if it’s a place location). In the case of a false positive, if it’s a place location this will result in an episode failure. If it’s a pick location and we have more pick attempts remaining then we mark that object as one we’ve already tried to pick and exclude it from further consideration. It could be possible for the uncertainty-weighted object score to be just below the threshold for the robot to pick/place on it. However after obtaining more views either the object will be detected (in which case the robot will move onto picking/placing on it) or the uncertainty will increase due to conflicting observations. There is some visualization of the change of uncertainty in the simulation examples the video provided in our supplement (after the hardware videos).

---

> > ### Author Response · Authors · 2025-11-21
> > **Q3**
> >
> > **Q3**. Imitation learning is a non-standard way to think about search tasks because ideally you would need to collect examples of search behavior, not just the path to complete the task. Consider the case that we just train to imitate a trajectory which goes directly to the target object. Then we are trying to directly learn the distribution of target objects instead of how to search pick locations for target objects. If the target objects are not in semantically meaningful locations or their locations are subject to domain shift (as may be the case because these are small, movable objects) then learning this distribution in the training data will not help us much outside of training. When performing search tasks for objects which are expected to be in semantically meaningful locations this approach is more viable. However if the robot operates in an environment where the object distribution is different to what it was trained on then it cannot be expected to do well due to the domain shift. Thus to create a system which we believe is capable of working in diverse real-world environments for this task, we think that our approach of leveraging foundation models which have been trained on large amounts of real-world data and using general principles of algorithm design which are not learnt from data is more tractable than IL.
> >
> > For reinforcement learning more generally, some of the baselines of the OVMM benchmark use RL [1]. You can see in table 1 that the baselines which use RL for the find object skill perform worse than those using the OVMM benchmark’s heuristic method for this skill. This could be because it can be difficult to define good intermediate rewards for RL systems performing search. Metrics like distance to the object would incentivise the system to try to learn the distribution of target object locations, which is undesirable as discussed above. As the episodes for this task can be long (1250 steps) and the success rate is not very high, without intermediate rewards it may be difficult for an RL system to learn how to do this task.
> >
> > [1] Yenamandra, S., Ramachandran, A., Yadav, K., Wang, A., Khanna, M., Gervet, T., ... & Paxton, C. (2023). Homerobot: Open-vocabulary mobile manipulation. NeurIPS

---

> > ### Author Response · Authors · 2025-11-23
> > **Sensitivity analysis complete**
> >
> > Please see Section 6.6 for the sensitivity analysis, we have filled in the rows which were blank in the previous revision now that those experiments have completed.

---

### Official Review · Reviewer_Yd22 · 2025-11-02

**Soundness:** 3
**Presentation:** 3
**Contribution:** 2
**Rating:** 4
**Confidence:** 2

**Summary:**

This paper present HELIOS, which is a hierarchical framework for language-specified mobile manipulation. The motivation of HELIOS is to address how to ground high-level language instructions into low-level visual representations when the environment is partially observable to the robot. It introduces a two-level scene representation for effective object search that integrates:
- a 2D semantic occupacy map
- a 3D Gaussian-based object representation for modeling task-relevant entities

The proposed framework is tested on OVMM benchmark in Habitat and it achieves the state-of-the-art performance compared to the baselines. Moreover, authors also test their method's performance on a Spot Robot of Boston Dynamics.

**Strengths:**

- Clear problem motivations: object search in partially observable environment is a very important problem in mobile manipulation
- Solid representation structure: combining 2-D semantic and occupancy map and 3D Gaussian splatting seems very reasonable
- Balanced exploration and exploitation object search: combining exploration and exploitation objective in search functions is reasonable
- Real-robot evaluation: authors not only show the validness of their method on simulation, but also validate the effectiveness on real-robot: Spot

**Weaknesses:**

- Limited novelty and lacking comparisons with existing works: Although it makes sense to combine 2-D semantic and occupancy map with 3D Gaussian splatting representations, all modules, e.g. semantic map, 3D Guassian representations, frontier exploration, uncertainty weighting is all existing techniques. It remains unclear to me what the novelty of the authors besides the combination of existing modules.
- Limited analysis on efficiency: authors claim the effectiveness of sparse 3DGS, however they lack the quatitative results comparisons of the FLOPS and parameters with baselines.
- Significantly lacking baseline comparisons: lacks quantitative comparisons with strong baselines. I am not familiar with this field, but I believe the literature is not vacant, I trust other reviewers on judging this.

**Questions:**

- How is the multi-view consistency constructed? Do you use probabilisty models?
- Have you compare against any open-world manipulation methods?
- How is language information grounded into the hierarchical representation?
- Could you provide quantitative results on inference efficiency and memory usage compared to dense scene graphs?

---

> ### Author Response · Authors · 2025-11-21
> **W1, W3, W2, Q1, Q2, Q3, Q4**
>
> Thank you for your detailed review. Please see our top-level comment for our response to the limited novelty concern of **W1**.
>
> **W1**, **W3**: This task requires the robot to be able to carry out many sub-skills: finding objects in a novel environment (itself an active research area called semantic object navigation or objectnav), navigating the robot base into a suitable location to grasp an object, navigating the robot base into a suitable location to place an object and placing the object, all without colliding with the environment or causing objects to collide with the environment. Because of this and the fact that we evaluate on a fairly recent (2023) benchmark [1] there are not many published works which we can easily compare to. While it would be great to compare our performance to more baselines it is not necessarily straightforward to transfer existing methods which do similar but not identical tasks into this benchmark due to the requirement of nontrivially extending the methods with new capabilities. There are methods which do sub-parts of this task (e.g. semantic object navigation or open-vocabulary table-top manipulation) but in order to use them for this task we would need to implement a mobile manipulation pipeline of which these methods would be a component rather than run just an already-existing method.
>
> **W2**: When we say that sparse 3DGS is efficient we mean in comparison to mapping the entire scene with 3D Gaussians. We have updated section 6.2 of our appendix to include a graph showing the average allocated and reserved GPU memory when running HELIOS in the form we propose in our paper with 3DGS used only to model the objects and when we model the whole scene 3DGS without changing anything else about our method. We provide a plot over time to give a sense of the relationship between the memory requirements and the scale of the scene being mapped. We measure the allocated memory just after the scene update to give an indication of the requirements for just keeping the scene representation in GPU memory, and also give the reserved memory to give better indication of the peak memory usage such as during updates or Information Gain calculations. The maximum of the average allocated memory is 4.1GB for HELIOS and 8.9GB for representing the full scene with 3DGS. For the average reserved memory the maximum is 4.6GB for HELIOS and 24.3GB for the full scene. The difference in the reserved memory is much larger than the difference in the allocated memory but even just the allocated memory is doubled when representing the full scene.
>
> **Q1**. The multi-view consistency is based on the number of times an object detector classifies pixels as belonging to an object of a certain class in observations taken during the episode. Please see the Preliminaries – Semantic classes for 3D Gaussian representation part of Section 3.1.3. In our paper.
>
> **Q2**. We do not compare to any open-world manipulation methods. Many of these methods are limited to table-top manipulation [2-4] and even those that are not often focus on manipulation skills such as grasping [5]. In contrast, our method uses existing methods for grasping and focuses on the high-level semantic reasoning aspects of this task. The simulation benchmark we evaluate our method in does not model the physics of grasping objects so it would be an unfair comparison for these methods because it is removing the impact of one of their core contributions. It is also difficult to transfer open-world manipulation methods which directly output actions [2,5] to other robot embodiments including in simulation.
>
> **Q3**. There are two ways in which language information is grounded into our hierarchical representation. The first is for the 2D part of our representation where we associate each observed image with a semantic similarity to the phrase “Seems like there is a (object) ahead”, obtaining a similarity for each task-relevant object in this way. We ground these similarities in our 2D layered value maps as described in Section 3.1.2. of our paper. This part is used to help to search for objects which have not yet been observed. The second part is that we ground detections made with an open-vocabulary object detector to the 3D part of our representation by modelling the distribution of semantic classes. Please see the Preliminaries – Semantic classes for 3D Gaussian representation part of Section 3.1.3. In our paper.
>
> **Q4**. Could you please clarify what you mean by “compared to dense scene graphs”? We are not aware of any dense scene graph method or framework which is used for this task or similar tasks.

---

> ### Author Response · Authors · 2025-11-21
> **References**
>
> [1] Yenamandra, S., Ramachandran, A., Yadav, K., Wang, A., Khanna, M., Gervet, T., ... & Paxton, C. (2023). Homerobot: Open-vocabulary mobile manipulation. NeurIPS
>
> [2] Li, Y., Deng, Y., Zhang, J., Jang, J., Memmel, M., Yu, R., ... & Goyal, A. (2025). Hamster: Hierarchical action models for open-world robot manipulation. ICLR.
>
> [3] Liu, F., Fang, K., Abbeel, P., & Levine, S. (2024). Moka: Open-world robotic manipulation through mark-based visual prompting. RSS.
>
> [4] Zhu, Minjie, Yichen Zhu, Jinming Li, Zhongyi Zhou, Junjie Wen, Xiaoyu Liu, Chaomin Shen, Yaxin Peng, and Feifei Feng. "Objectvla: End-to-end open-world object manipulation without demonstration." arXiv preprint arXiv:2502.19250 (2025).
>
> [5] Stone, A., Xiao, T., Lu, Y., Gopalakrishnan, K., Lee, K. H., Vuong, Q., ... & Hausman, K. (2023). Open-world object manipulation using pre-trained vision-language models. arXiv preprint arXiv:2303.00905.

---

### Official Review · Reviewer_gr26 · 2025-11-03

**Soundness:** 3
**Presentation:** 3
**Contribution:** 2
**Rating:** 8
**Confidence:** 4

**Summary:**

In this work, the authors present a hierarchical scene representation with the goal of open-vocabulary mobile manipulation. The presented scene representation is composed of two tasks: the first being a 2D, birds eye view containing semantic, occupancy, and value information, and a 3DGS based volumetric map that keeps track of objects in the scene. The primary novelty in this work is the integration of this confidence aware 3DGS representation that prompts the agent to add more views of the target object or location if there is more uncertainity over the classification. The authors present evaluations of their method in the simulated OVMM benchmark, and compare it with baselines from HomeRobot and MoManipVLA. The authors also present the baselines, where they show that integrating the hierarchical structure and the confidence based methods help in each individual steps.

**Strengths:**

* integrating a 3D layer over the 2D layers for search and object discovery makes the scene representation more robust to multi-view inconsistencies, which is one of the primary cases where object recognition and discovery fails in OVMM.
* The uncertainty aware 3DGS representation gives a great way to both represent objects in the scene and also create a good way to resolve object detection noise.
* Improving the scene representation over multiple pick and places gives a good way to learn an online, dynamic representation of the map, which is necessary for the real world.

**Weaknesses:**

* The presented method works with a fixed number of pre-determined class, which belies the "open vocabulary" claim. While the authors claimed that we can pre-select a fixed number of classes, that does break the open vocabulary assumption in reality.
* The real world experiments should be more front-and-center than they are right now. As it is, it doesn't seem to be a big part of the submission where this work would truly shine with them.
* The paper does not give us enough detail about the local search process, and how it is performed with obstacles in the environment.

**Questions:**

* Why are real world experiments done with monocular depth estimation? If the representations are 3DGS based, should that not be sufficient to figure out a depth representation that works together with it?
* What are the experimental numbers from the real world trials?
* What are the primary causes of failure right now? An analysis similar to OK-Robot's waterfall chart would be very helpful to understand the current limitation.

---

> ### Author Response · Authors · 2025-11-21
> **W1, W3, Q1, Q3 (W2 and Q2 in top-level comment)**
>
> Thank you for your detailed review. Please see our top-level comment for our response about quantitative results in the real world (**W2**, **Q2**).
>
> **W1**. We consider our method to be open-vocabulary because the objects only need to be defined before a given task rather than being pre-defined for the method as a whole. It is true that our method cannot handle changing the objects referred to in the instruction mid-way through an episode, however we don’t believe that such a capability is a requirement in order to be considered open-vocabulary. We note that the benchmark we evaluate on has “open-vocabulary” in its name (“Homerobot: Open Vocabulary Mobile Manipulation” [1]), so we think this is a reasonable interpretation of the meaning of “open-vocabulary”.
>
> **W3**. Thank you for the feedback, we have added some more detail to the local search part of Section 3.2. to make it clear how we avoid obstacles around the object being searched.
>
> We do not currently take into account obstacles on the pick location that would block the target object itself from view. A possible extension to our work to address situations where there is a lot of clutter at the pick location would be to model all objects in this region with 3D Gaussians, as this could be used to determine viewing angles where the view of the target object is not blocked. Although this would only work in situations where the target object had been observed prior to the local search or if the gaze points were adaptively set during local search. Note that the class of these additional objects is unimportant to the task so they could be modelled as the “other” class rather than requiring their classes to be known before starting the episode. While this would increase the computational requirements it would still be less than modeling the entire scene with 3D Gaussians, and it would also be possible to delete the Gaussians representing the other objects after local search to further reduce the computational requirements. This extension is outside the scope of this rebuttal but would definitely be interesting to look into in the future.
>
> **Q1**. We use monocular depth estimation to better initialize the 3D Gaussians and also to provide them with another training signal while optimizing. Note that our views of the scene are obtained online, so when first seeing an object we need to initialize 3DGS from one view which without depth is an ill-posed problem.
>
> **Q3**. Please see Section 6.4 in the appendix of the updated paper. We did not do a manual assignment of failure causes due to the large number of episodes, instead using the per-episode metrics saved by the OVMM benchmark.
>
> [1] Yenamandra, S., Ramachandran, A., Yadav, K., Wang, A., Khanna, M., Gervet, T., ... & Paxton, C. (2023). Homerobot: Open-vocabulary mobile manipulation. NeurIPS

---

### Official Review · Reviewer_9iMV · 2025-11-03

**Soundness:** 3
**Presentation:** 2
**Contribution:** 2
**Rating:** 4
**Confidence:** 4

**Summary:**

This paper presents HELIOS, a hierarchical framework designed to address language-guided pick-and-place mobile manipulation tasks in open, partially observed scenes. It integrates 2D semantic/occupancy maps for efficient navigation and sparse 3D Gaussian representations for task-relevant object modeling, alongside a search objective that balances exploration of unobserved regions and exploitation of semantic cues. The work demonstrates performance on the OVMM benchmark (outperforming baselines like HomeRobot and MoManipVLA in limited/potential settings) and zero-shot transfer to a real-world Spot robot, with ablation studies verifying the utility of its hierarchical representation and uncertainty-weighted object scoring.

**Strengths:**

One potential strength of the work is its **practical integration of multi-modal scene representation for end-to-end language-guided pick-and-place tasks**, which bridges 2D semantic/occupancy maps (for efficient navigation) and sparse 3D Gaussians (for precise object modeling). Unlike some prior methods that rely solely on 2D maps (e.g., HomeRobot) or dense 3D scene representations (e.g., early 3D Gaussian-based SLAM works), HELIOS tailors its layered structure to the specific needs of mobile manipulation—using 2D maps to prioritize exploration and 3D Gaussians to track task-relevant objects with multi-view consistency. This design, while building on existing components, aligns representation granularity with real-world manipulation demands (e.g., avoiding redundant full-scene 3D modeling) and enables direct zero-shot transfer to physical robots (Spot) without additional data, demonstrating a degree of engineering utility for practical deployment scenarios.

**Weaknesses:**

1. **Weak Motivation Grounding**: The paper fails to clearly pinpoint unaddressed gaps in existing literature. It only lists generic challenges (partial observation, semantic grounding) but does not quantify or exemplify how state-of-the-art methods (e.g., HomeRobot, MoManipVLA) struggle with these issues—for instance, no data on prior methods’ false detection rates or inefficient exploration is provided. This makes the necessity of HELIOS unconvincing, as it does not solve a well-documented, unmet need.

2. **Lack of Methodological Novelty**: Core components are incremental tweaks of existing techniques rather than original innovations. The 2D semantic value map extends Yokoyama et al. (2023b) with minor multi-target support; the sparse 3D Gaussian object modeling adapts Lu et al. (2024)’s 3D Gaussian framework without new modeling logic; and the “exploration-exploitation balance” relies on classic information gain calculations, with no novel objective function design. There is no theoretical or algorithmic breakthrough to advance the field.

3. **Insufficient Experimental Validation**: While the paper claims state-of-the-art results on OVMM, key comparisons are incomplete. For example, real-world Spot robot experiments lack side-by-side performance metrics (e.g., task success time, grasp failure rate) with other zero-shot manipulation methods. Additionally, the “place skill” is acknowledged as a major failure source, but no analysis of why HELIOS cannot address this (or how it compares to methods with better placing) is provided, weakening the work’s practical credibility.

4. **Unconvincing Zero-Shot Transfer Claim**: The paper asserts zero-shot transfer to real-world environments but provides limited details to support this. There is no analysis of how HELIOS handles real-world noise (e.g., variable lighting, occlusions) that differs from simulation, nor is there evidence that its performance in the office setting is superior to or even on par with existing zero-shot manipulation systems. The transferability claim thus remains unsubstantiated.

**Questions:**

1. The paper mentions that HELIOS uses sparse 3D Gaussians to model task-relevant objects instead of dense full-scene 3D representations. However, it does not clarify: how does the framework **dynamically define "task-relevant objects"** from ambiguous language instructions (e.g., instructions like "fetch the small item on the shelf")? Is there a risk of missing key objects due to over-sparsification, and how is this risk mitigated?

2. In the real-world Spot robot experiments, the paper claims "zero-shot transfer" but lacks comparative data with SOTA zero-shot manipulation methods (e.g., HomeRobot’s real-world variants). Could you provide **quantitative metrics** (e.g., task success rate, average execution time, grasp failure rate) that directly compare HELIOS with these methods in the same office environment?

3. The discussion notes that HELIOS occasionally fails at the "place" step due to improper gripper angle adjustment. Unlike the "pick" step (supported by multi-view 3D Gaussian verification), the paper does not explain: why is there no similar **3D geometric verification mechanism** for the "place" target (e.g., checking if the object aligns with the placement surface)? Is this a deliberate design choice, or a limitation to be addressed?

4. The 2D semantic value map in HELIOS extends Yokoyama et al. (2023b) to multi-target scenarios. However, it is unclear: when multiple targets overlap in the 2D map (e.g., two cups on a table viewed from above), how does the framework **resolve spatial ambiguity** to avoid guiding the robot to the wrong target? What modifications were made to the original single-target value map logic to handle this?

5. The paper states that HELIOS balances exploration and exploitation via "information gain" calculations. But in cluttered environments (e.g., a desk with many small objects), the "information gain" of unobserved regions may be low. How does the framework **prevent stagnation** (e.g., the robot repeatedly checking the same high-probability area without exploring new regions) in such cases?

---

> ### Author Response · Authors · 2025-11-21
> **W1, W3, Q3, Q1 (W2, W4 and Q2 in top-level comment)**
>
> Thank you for your detailed review. Please see our top-level official comment for detailed responses to **W2**, **W4** and **Q2**. **W3** is also mostly covered by this comment but we will provide some additional details here.
>
> **W1**. We discuss this gap in the literature and how our method addresses it in our related work section: “benchmarks for targeted instantiations of this problem have identified that a major cause of failure in this task is correctly finding and identifying objects for performing pick and place”. We did not also include this motivation in the introduction due to space constraints but we agree that including it in the introduction would strengthen the motivation for our method. Thank you for pointing this out. We have updated the paper with this change, please see the introduction in the revised version (starting with “In particular, incorrect object detections”).
>
> **W3**, **Q3**. In terms of the place skill, while this is a necessary component of a mobile pick-and-place method our work focuses on other challenging perception and reasoning aspects of this task. For contextual information, in our opinion, force-informed solutions to manipulation (i.e. dynamic trajectory planning), would yield the best solutions to effective manipulation subskills. This research philosophy drove our decision to build reasoning algorithms over 3D object models which could directly improve results from such a policy. Thus, unfortunately, we consider improving the place skill outside of the scope of our research in this paper, in particular because Habitat does not simulate the physics in a manner which allows for force-informed solutions to manipulation with contact.
>
> Because our method is modular it is possible to swap just this component of our method when a better place skill is developed. As using 3D Gaussians to represent objects for manipulation is an active area of research [1-3], it may be possible to combine these with our approach using our scene representation or a slightly modified version of it. However, to effectively test such a solution we would need to use a simulator with higher fidelity physics modeling.
>
> **Q1**. Right now, for the instructions we deal with in this work the relevant object classes are always provided. For more complicated language, we would first try using an LLM to extract the relevant objects. For cases like the example you gave where the object relationships are very important to defining the target object, we would probably need to enforce a spatial relationship between objects before grasping. This would require an extension of our planning to include this constraint. For object classes which are ambiguous, such as “small object” in your example or demand-driven navigation targets such as “something to drink”, the simplest way to deal with this is to model all objects which meet this criteria as being in the same class for the purposes of our 3D semantic representation. Then the question is whether there is an object detector which is capable of detecting the ambiguous class. If not, one approach would be to ask an LLM to list all object classes which could fit the criteria of the ambiguous class but this may result in missing relevant object classes as you suggest. Another approach would be to ask a VLM if such an object is present in the image. If the object is present to obtain the segmentation mask we can either get its class from the VLM and use an object detector or get the pixel location from the VLM (methods like ViGoRL [4] can allow VLMs to provide this information more reliably) and use SAM [5-7] with that point as the prompt.

---

> ### Author Response · Authors · 2025-11-21
> **Q4, Q5**
>
> **Q4**. The 2D value maps are only used for exploration as they indicate likely areas to find the object(s) we are looking for. It does not matter if objects of the same class overlap on them because we only model similarity scores per RGB image which are aggregated in a heatmap, not object level representations. We use the 3D part of our scene representation which models instances to represent the task-relevant objects. If you mean when two objects of different classes which we are looking for overlap, note that we keep separate map layers for each task-relevant object class (as pictured in fig 1 and discussed in Sec 3.1.2). Thus we can simply use the layer which corresponds to the object we are currently looking for.
>
> **Q5**. In practice, because we generate set viewpoints we mark visited start receptacles and don’t revisit them (please see the local search section of Sec 3.2). Note that even without this, our global objective score for objects multiplies the class score of the object with the information gain so even if the class score is high a low information gain will reduce the global objective score significantly, especially keeping in mind that the maximum class score is 1.
>
> [1] Shorinwa, O., Tucker, J., Smith, A., Swann, A., Chen, T., Firoozi, R., ... & Schwager, M. (2024). Splat-mover: Multi-stage, open-vocabulary robotic manipulation via editable gaussian splatting. CoRL
>
> [2] Zheng, Y., Chen, X., Zheng, Y., Gu, S., Yang, R., Jin, B., ... & Wang, M. (2024). Gaussiangrasper: 3d language gaussian splatting for open-vocabulary robotic grasping. IEEE Robotics and Automation Letters.
>
> [3] Yu, J., Hari, K., El-Refai, K., Dalal, A., Kerr, J., Kim, C. M., ... & Goldberg, K. (2025). Persistent object gaussian splat (pogs) for tracking human and robot manipulation of irregularly shaped objects. arXiv preprint arXiv:2503.05189.
>
> [4] Sarch, G., Saha, S., Khandelwal, N., Jain, A., Tarr, M. J., Kumar, A., & Fragkiadaki, K. (2025). Grounded Reinforcement Learning for Visual Reasoning. arXiv preprint arXiv:2505.23678.
>
> [5] Kirillov, A., Mintun, E., Ravi, N., Mao, H., Rolland, C., Gustafson, L., ... & Girshick, R. (2023). Segment anything. In Proceedings of the IEEE/CVF international conference on computer vision (pp. 4015-4026).
>
> [6] Ravi, N., Gabeur, V., Hu, Y. T., Hu, R., Ryali, C., Ma, T., ... & Feichtenhofer, C. (2024). Sam 2: Segment anything in images and videos. arXiv preprint arXiv:2408.00714.
>
> [7] Carion, N., Gustafson, L., Hu, Y. T., Debnath, S., …&  Feichtenhofer, C. (2025) SAM 3: Segment Anything with Concepts

---

### Official Review · Reviewer_5ipo · 2025-11-10

**Soundness:** 3
**Presentation:** 3
**Contribution:** 2
**Rating:** 4
**Confidence:** 3

**Summary:**

This paper addresses open-vocabulary mobile manipulation in partially observable indoor environments. The authors propose HELIOS, a hierarchical exploration system that integrates navigation, perception, and manipulation through a three-layer scene representation. The first layer is a 2D occupancy map for navigation, the second is a semantic value map for prioritizing frontiers, and the third is a sparse, task-relevant 3D Gaussian representation that maintains Dirichlet statistics for language-grounded semantic consistency. The key contribution is a unified global objective function that balances exploration (visiting previously unseen frontiers) and exploitation (re-observing previously detected objects to reduce uncertainty), utilizing semantic scores, information gain, and distance penalties. The method is evaluated on the Habitat OVMM benchmark and demonstrates improved success rates in finding and manipulating objects compared to baseline systems.

**Strengths:**

1.	The paper tackles a real gap in open-vocabulary mobile manipulation — the high failure rate in locating objects or placing them correctly due to unstable open-vocabulary perception. By explicitly modeling multi-view semantic consistency, the method addresses this critical bottleneck.
2.	The hierarchical representation design is coherent: metric navigation, semantic reasoning, and object-level language understanding are neatly decoupled yet connected through a unified planning objective. This makes the system modular and interpretable. The use of information gain and uncertainty reduction in the unified objective provides a principled mechanism for switching between exploration and local observation, rather than relying on heuristic thresholds or stage-wise rules.
3.	The experiments follow a standard benchmark (HomeRobot/OVMM) with clear stage metrics (FindObj, Pick, FindRec, Place, SR), demonstrating reproducibility and measurable improvements.

**Weaknesses:**

1.	The claimed novelty overlaps with recent work from 2024–2025 on language-embedded 3D Gaussian representations and task-driven active exploration. In particular, ATLAS Navigator (2025) already integrates language-conditioned 3D Gaussians with information gain–based viewpoint selection for task-driven exploration. The current paper should more clearly delineate how its global objective and sparse Gaussian representation differ from ATLAS and similar approaches.
2.	Recent methods such as Splat-MOVER, GaussianGrasper, and POGS already apply object-level Gaussian representations for open-vocabulary manipulation. The present paper’s claim that it is the first to use online, task-relevant Gaussian maps for manipulation is therefore too strong. A quantitative or qualitative comparison to these systems would strengthen the paper’s contribution statement.
3.	The proposed “frontier vs. re-observation” trade-off has been previously explored in hierarchical semantic navigation papers (e.g., “Uncertainty-Aware Embodied Exploration,” 2024). The paper should emphasize that its novelty lies in embedding both behaviors into a single scalar objective, rather than fixed scheduling, and provide ablation evidence to support this claim.
4.	While the results show performance gains, it remains unclear how much of the improvement is due to the 3D Gaussian consistency versus the global objective design. Further analysis isolating these effects would clarify the main source of success.

**Questions:**

1.	How exactly does your unified global objective differ mathematically from ATLAS Navigator’s task-driven information-gain formulation?
2.	In the Dirichlet update of object semantics, how do you avoid bias toward frequently observed (but incorrect) objects, e.g., normalizing by observation count or adding temporal decay?

---

> ### Author Response · Authors · 2025-11-21
> **W1, Q1, W2**
>
> Thank you for your detailed comments and providing many relevant references.
>
> **W1** and **Q1**: For ATLAS navigator specifically, as far as we can tell they do not use information gain in their objective at all. They also seek to create an efficient hierarchical scene representation for navigating in unseen environments, although their approach is to do dense mapping with 3DGS and then compose this into submaps which can be loaded in and out of active memory. Through their hierarchical structure, high-level planning can occur on the level of sub-maps which prevents them from needing to consider all the Gaussians during planning. However they still need to create and maintain dense 3DGS maps for their method, while these can be loaded out of active memory they still need to be stored somewhere and this can incur time costs especially if the maps are not stored on-board. If you are doing a task which requires a dense 3D map this can make sense, in their case they use this dense map for collision avoidance which may be relevant for planning in 3D for UAVs or in situations where the robot needs to go under obstacles. We argue that this dense 3D representation is not required for the mobile pick and place tasks we address in our work and so instead of constructing a sparse abstraction on top of a dense 3D representation we directly construct a sparse 3D representation.
>
> In terms of the mathematical differences between our objective functions, they have two objective functions: one for high-level selection of waypoints and one for continuous trajectory planning with collision avoidance. The first is more comparable to our objective function. According to eq 7 they are trying to maximize the utility of the visited vertices subject to the constraint of staying within a given travel budget. We could not find a precise definition of their utility, but it seems to be the semantic similarity between the object-level cluster of Gaussians to the task-relevant terms (based on “From these clusters, we propagate the object-level relevancy up the tree to obtain utility scores for region- and submap-level nodes” and “We can then recover the original dimensions of the features with the inverse of the PCA for computing similarity with the task embeddings.”). In this case, it is similar to the score from our value map.
>
> **W2**: We do not claim that we are the “first to use online, task-relevant Gaussian maps for manipulation”. If there is a section of our work which you believe is stating this claim please let us know where it is so we can edit for clarity. As for the question of comparing to previous work performing open-vocabulary robotic manipulation using 3DGS, we would like to point out that we are addressing mobile manipulation whereas these works are table-top manipulation. While there is some overlap in challenges between these tasks and the skills required to complete them, there are also challenges distinct to mobile manipulation, and these aspects of the problem are what we are focused on. Efficiently searching a large spatial environment is a key challenge in the mobile pick-and-place tasks in novel environments which we address in our work, a challenge which is not present in table-top manipulation tasks. Our method does not innovate in the actual grasping skill itself but rather focuses on efficiently finding the target object while taking into account the fact that the object detector it uses is imperfect. It is also generally not possible to directly run a table-top manipulation method for mobile manipulation, so we would need to implement a mobile manipulation pipeline of which these methods would be a component rather than run just an already-existing method.

---

> > ### Author Response · Authors · 2025-11-21
> > **W3, W4, Q2**
> >
> > **W3**. We could not find the “Uncertainty-Aware Embodied Exploration,” 2024 paper you provided as a reference. Did you mean “Embodied Uncertainty-Aware Object Segmentation” [1]? If not could you please provide some more details so we can locate the paper?
> >
> > We feel that we already claim as a contribution our specific objective function and its integration with our scene representation rather than the idea of an exploration vs exploitation tradeoff itself (e.g. from our contributions: “an objective function on our hierarchical scene representation that balances exploring the scene to find regions which might contain the target object with exploiting observed semantic information”). We note that we have already done an ablation study of removing our global search objective and instead always performing local search of each candidate object before going to frontiers (W/o global search objective ablation). If there are any specific sections which you feel gave the impression that we were claiming the idea of an “explore-exploit” or “frontier vs. re-observation” trade-off was itself novel, it would be very helpful if you could point them out so that we can re-write them to clarify our claims.
> >
> > **W4**. Please see the ablation study in Table 1 of our paper, we include an ablation of our method removing the global search objective (W/o global search objective ablation).
> >
> > **Q2**: The observation count is taken into account when calculating the expected value or uncertainty from the concentration parameters (please see Eq 4). These equations normalize using the sum of the concentration parameters of all the classes, which is a form of normalizing by the observation count.
> >
> > [1] Fang, X., Kaelbling, L. P., & Lozano-Pérez, T. (2024, October). Embodied Uncertainty-Aware Object Segmentation. In 2024 IEEE/RSJ International Conference on Intelligent Robots and Systems (IROS) (pp. 2639-2646). IEEE.

---

### Author Response · Authors · 2025-11-21
**Overview and list of changes**

We would like to thank all of the reviewers for their detailed and helpful feedback. Many reviewers recognized the importance of the problem we address in this work, stating that it “tackles a real gap in open-vocabulary mobile manipulation” (5ipo), “[has c]lear problem motivations” (Yd22), and “aims at addressing a pressing open problem in embodied AI” (JyoA). Most reviewers also found our representation suitable for the task, 9iMV notes that “HELIOS tailors its layered structure to the specific needs of mobile manipulation” and “aligns representation granularity with real-world manipulation demands”, 5ipo describes it as “coherent… modular and interpretable” and Yd22 as “solid… very reasonable”, and JyoA states that “[t]he hierarchical integration of 2D and sparse 3D Gaussian maps for exploration and manipulation is novel and elegantly addresses multi-scale reasoning”. Likewise many reviewers described our global objective function in a positive light as “principled” (5ipo, JyoA) or “well-formulated” (UXNY). UXNY describes our experiments as “[c]omprehensive… [i]ncludes comparisons against strong baselines…, ablation studies, limited vs. unlimited pick attempts, and real-world robot demonstrations”. Aspects of this were also reflected in other reviews, with jyoA describing our ablation studies as “extensive and well-designed to isolate each contribution” and stating that “[c]lear experimental validation shows consistent improvement over strong baselines”, 5ipo notes that our experiments “follow a standard benchmark… with clear stage metrics… demonstrating reproducibility and measurable improvements”, and Yd22 states “authors not only show the validness of their method on simulation, but also validate the effectiveness on real-robot”.

We will address some of the common feedback here to avoid needing to repeat points across different responses. We also provide a list of changes in the updated version of the paper here.

List of changes to paper in latest version:
* Abstract: re-word last sentence (now last two sentences) to make clear that we are not claiming “zero-shot transfer”
* Introduction: Add motivation for our specific approach following feedback from reviewer 9iMV (changes start at “In particular, incorrect object detections…” and go until the start of the contributions)
* Contributions: re-word last sentence to only state what we actually did without mentioning “zero-shot” or “transfer”.
* Local search section of method: Add additional details following feedback from reviewer gr26
* Figures 2 and 3: re-format to use full width of paper following feedback from reviewer wuZW
* Appendix – Section 6.2: re-word to make clear our method runs on one GPU at a time not across 8 GPU nodes (apologies for the misleading wording). Add GPU memory usage comparison for our method versus representing the entire scene with 3D Gaussians following feedback from reviewer Yd22.
* Appendix – Section 6.4: add failure-cases diagrams following feedback from reviewer gr26
* Appendix – Section 6.5: add a small ablation study on the effect of changing the assumption used to calculate the Information Gain following feedback from reviewer UXNY
* Appendix – Section 6.7: add a sensitivity analysis following feedback from reviewer JyoA.

---

> ### Author Response · Authors · 2025-11-21
> **Novelty and contribution**
>
> A weakness listed by multiple reviewers (9iMV W2, Yd22 W1, JyoA W1, wuZW W1, UXNY W1) is that key components of our method are obtained from prior work and that our method does not contain fundamentally new algorithms. However, constructing a useful representation for a task from prior work addressing a different task or with foundation models is a valuable contribution to ICLR and related communities. The fact that we can formulate a new task objective over these combined representations **without requiring training** is a mark of the power of our proposed representation. We show the significance of leveraging common representations from other robotics tasks to address new ones by performing object goal navigation in addition to open vocabulary pick and place using the same layered representation and no training. We also note that, as illustrated earlier by quotes from the reviews, many reviewers found the design of our representation and objective function suitable for the task. Thus the primary concern of most reviewers seems to us to be about the novelty and contribution of a representation which is built using existing components from prior work combined with a planning algorithm which uses classical theory, rather than concerns about how suitable our representation or our objective function is to the task we address or related tasks.
>
> Many works on lifting features from 2D foundation models to 3D representations such as 3DGS and the applications this enables have been published in ICLR and related conferences [1-7]. The core components of these methods are also prior work, with innovations in how they fuse the information from these prior methods into a unified representation suitable for their applications. We argue that fusing information from our 2D value map representation and sparse 3D semantic representation is a similar type of innovation. While we make use of prior work [8] on semantic multi-view consistency for the 3D part of our representation, we note that this work did not model instances, represented the entire scene with 3D Gaussians, and was used to represent semantic information from a closed-set semantic segmentation network without showing how this could be extended to open-vocabulary tasks. We also point out that to the best of our knowledge prior work which models individual objects with 3DGS also models the background with 3DGS [9-12], likely because these works have largely been used in tabletop manipulation settings where the spatial scale of the scene is much lower. While it should be straightforward to use these works to extract and store 3D Gaussians only for the objects, we note this leaves the representation missing geometric information required for things like path planning which we solve in our representation by also including 2D maps which store this information. As using 3DGS representations for robotics and embodied AI tasks is a thriving research area, we think that proposing a way to increase the efficiency of these representations for object-centric mobile tasks is a valuable contribution.

---

> > ### Author Response · Authors · 2025-11-21
> > **Hardware part 1**
> >
> > Another concern or question of multiple reviewers (9iMV W3 W4 Q2, gr26 Q2, wuZW W3) was the lack of quantitative results for our hardware demonstrations. The first point we need to address is our usage of the words “zero-shot” and “transfer”. We apologize for the confusion we caused by using the word “transfer”. Our intention was to state that our method is zero-shot and that we ran a demonstration in the real world without needing to obtain and train on any data. By “our method is zero-shot” we mean that we didn’t train anything for this specific task, and all our pretrained components were foundation models actually trained on real world data in the first place. We initially thought our phrasing of “As our approach is zero-shot, HELIOS can also transfer to the real world, as we illustrate by demonstrating it in a real world office environment on a Spot robot.” made this distinction.. However, this phrasing is misleading because we did not train in simulation and so running the method in the real-world is not “transfer”. We have reworded the last sentence of our abstract and our contributions to remove the word “transfer” and make our intended meaning clearer. The phrasing is now “We also demonstrate HELIOS performing language specified pick and place in a real world office environment on a Spot robot. Our method leverages pretrained VLMs to achieve these results in simulation and the real world without any task specific training.”
> >
> > Next, we would like to point out that we only ever claim to perform hardware demonstrations not experiments and to clarify what we view as the difference. Demonstrations only show that the system is capable of doing something, it does not make a claim about generalizability or overall performance. In order to make such claims, experiments are required, and these should include enough trials to have statistically significant findings, vary factors which the method is expected to be robust to and ideally include comparisons to other methods. We perform these sorts of experiments purely in simulation because we consider the scale of experiments required to make claims about the generalizability and overall performance of tasks involving long-horizon navigation, such as the mobile pick-and-place task we address in this paper, to be intractable in an academic research setting. While some works that involve long horizon navigation provide quantitative results on hardware for a few scenarios as part of their experiments, they are not at the same scale as can be done in simulation both in terms of the number of trials and the scale and diversity of the environments. For example, thousands of episodes are evaluated in simulation versus often less than 100 in the real world, and real-world experiments often take place in the same building and/or are limited to room-scale instead of house-scale. There are works which perform large-scale hardware experiments but in this case the experiments generally form a large part of their contribution because of the required monetary and time investment. E.g. Gervet et al [13] conduct experiments using existing methods for semantic object navigation in the real world and their key finding is about the performance of different types of methods in the real world versus in simulation. OK-robot [14] perform significant experiments in the real world for a similar task to us, although these experiments are limited to room-scale. They describe their work as a “systems-first approach” rather than arguing for the novelty of the method. A key reason why embodied AI navigation benchmarks in simulation are created and are valuable is because they allow us to make statements about task performance and show generalizability w.r.t. some factors (e.g. different environment layouts) without needing to perform these costly experiments in the real world. They also provide a standard evaluation procedure which allows for a fair comparison between methods, which is much harder to do in the real world.
> >
> > While it is true that the simulators these benchmarks are created in are not a perfect representation of reality and so they are not always indicative of real world performance, this discrepancy is most likely to occur when methods are trained in the simulator and thus fit to a distribution which is different to what they would encounter when doing that task in the real world.  Both low level actions and sensor data such as RGB-D images are not perfectly simulated in any software. We emphasize the zero-shot nature of our method because all learnt components we use are trained on large amounts of real-world data instead of simulation data, so we expect to mostly circumvent these issues.

---

> > > ### Author Response · Authors · 2025-11-21
> > > **Hardware part 2**
> > >
> > > In addition, our method focuses on high-level planning and uses separate low-level controllers for sub-skills like navigating to waypoints and grasping objects, impacting the comparison of our method in simulation versus the real world. We can take advantage of mature robotics techniques including those available through the APIs of commercial robotics systems like Spot to do these subskills in the real world instead of relying on low-level actions trained in an environment with imperfect physics simulation. Note that this is not the case for many table-top manipulation tasks which try to learn action policies in simulation and then have a physics-driven sim-to-real gap in the execution of the action policy. These methods, therefore, often need to be evaluated in the real world as the discrepancies between the physics of the simulator and the real world have a large impact on the performance of core components of their method. It is also more tractable to sufficiently vary relevant factors in table top manipulation as they operate in a much smaller spatial environment.
> > >
> > > For navigation methods which evaluate on simulation benchmarks, hardware demonstrations like the ones we provide in this work are often considered sufficient evidence to claim viability in real-world scenarios [15, 16, 17]. Of course we do agree with the reviewers that our paper could be strengthened by performing hardware experiments and reporting quantitative results. However, key characteristics  of our method address the problem of building scale semantic reasoning which requires testing in more than one building to sufficiently validate a meaningful success rate, a large scale experimental effort. Our simulation experiments derive a meaningful understanding of success rate over spatial-semantic variation, but our hardware demonstrations show computational viability of replicating these results across diverse environments given a large scale deployment effort.

---

> ### Author Response · Authors · 2025-11-21
> **References**
>
> Thank you again to all the reviewers for your suggestions in improving our paper!
>
> References:
>
> [1] Peng, Q., Planche, B., Gao, Z., Zheng, M., Choudhuri, A., Chen, T., ... & Wu, Z. (2024). 3d vision-language gaussian splatting. ICLR
>
> [2] Zhang, C., & Lee, G. H. (2025). econSG: Efficient and Multi-view Consistent Open-Vocabulary 3D Semantic Gaussians. ICLR
>
> [3] Zhou, S., Chang, H., Jiang, S., Fan, Z., Zhu, Z., Xu, D., ... & Kadambi, A. (2024). Feature 3dgs: Supercharging 3d gaussian splatting to enable distilled feature fields. In Proceedings of the IEEE/CVF Conference on Computer Vision and Pattern Recognition (pp. 21676-21685).
>
> [4] Ye, J., Wang, N., & Wang, X. (2023). Featurenerf: Learning generalizable nerfs by distilling foundation models. In Proceedings of the IEEE/CVF International Conference on Computer Vision (pp. 8962-8973).
>
> [5] Kerr, J., Kim, C. M., Goldberg, K., Kanazawa, A., & Tancik, M. (2023). Lerf: Language embedded radiance fields. In Proceedings of the IEEE/CVF international conference on computer vision (pp. 19729-19739).
>
> [6] Qin, M., Li, W., Zhou, J., Wang, H., & Pfister, H. (2024). Langsplat: 3d language gaussian splatting. In Proceedings of the IEEE/CVF Conference on Computer Vision and Pattern Recognition (pp. 20051-20060).
>
> [7] Kobayashi, S., Matsumoto, E., & Sitzmann, V. (2022). Decomposing nerf for editing via feature field distillation. Advances in neural information processing systems, 35, 23311-23330.
>
> [8] Wilson, J., Almeida, M., Sun, M., Mahajan, S., Ghaffari, M., Ewen, P., ... & Sen, A. (2025, May). Modeling uncertainty in 3d gaussian splatting through continuous semantic splatting. In 2025 IEEE International Conference on Robotics and Automation (ICRA) (pp. 3284-3290). IEEE.
>
> [9] Lu, Y., Zhou, Y., Qiao, Y., Song, C., Liang, T., Ma, J., ... & Yin, Y. Segment then Splat: Unified 3D Open-Vocabulary Segmentation via Gaussian Splatting. In The Thirty-ninth Annual Conference on Neural Information Processing Systems.
>
> [10] Shorinwa, O., Tucker, J., Smith, A., Swann, A., Chen, T., Firoozi, R., ... & Schwager, M. (2024). Splat-mover: Multi-stage, open-vocabulary robotic manipulation via editable gaussian splatting. CoRL
>
> [11] Zheng, Y., Chen, X., Zheng, Y., Gu, S., Yang, R., Jin, B., ... & Wang, M. (2024). Gaussiangrasper: 3d language gaussian splatting for open-vocabulary robotic grasping. IEEE Robotics and Automation Letters.
>
> [12] Yu, J., Hari, K., El-Refai, K., Dalal, A., Kerr, J., Kim, C. M., ... & Goldberg, K. (2025). Persistent object gaussian splat (pogs) for tracking human and robot manipulation of irregularly shaped objects. arXiv preprint arXiv:2503.05189.
>
> [13] Gervet, T., Chintala, S., Batra, D., Malik, J., & Chaplot, D. S. (2023). Navigating to objects in the real world. Science Robotics, 8(79), eadf6991.
>
> [14] Liu, P., Orru, Y., Vakil, J., Paxton, C., Shafiullah, N. M. M., & Pinto, L. (2024). Ok-robot: What really matters in integrating open-knowledge models for robotics. RSS
>
> [15] Yokoyama, N., Ha, S., Batra, D., Wang, J., & Bucher, B. (2024, May). Vlfm: Vision-language frontier maps for zero-shot semantic navigation. In 2024 IEEE International Conference on Robotics and Automation (ICRA) (pp. 42-48). IEEE.
>
> [16] Yu, B., Kasaei, H., & Cao, M. (2023, October). L3mvn: Leveraging large language models for visual target navigation. In 2023 IEEE/RSJ International Conference on Intelligent Robots and Systems (IROS) (pp. 3554-3560). IEEE.
>
> [17] Yuxuan Kuang, Hai Lin, and Meng Jiang. 2024. OpenFMNav: Towards Open-Set Zero-Shot Object Navigation via Vision-Language Foundation Models. In Findings of the Association for Computational Linguistics: NAACL 2024, pages 338–351, Mexico City, Mexico. Association for Computational Linguistics.

---

### Author Response · Authors · 2025-11-23
**Paper update: sensitivity analysis now complete**

The experiments for the sensitivity analysis have finished running and the paper has been updated with these numbers. The list of changes in the previous review has been updated. No further changes to the revised paper are planned until after reviewers respond to comments.

---

### Meta-Review · Area_Chair_5L8A · 2025-12-21

**Summary:**

The paper received divergent ratings from the reviewers (8,6,6,4,4,4,2). While the reviewers appreciated strengths such as tackling a real gap in open-vocabulary mobile manipulation, clear motivation, and achieving improvements over strong baselines, they raised several concerns such as:

(a) Lack of novelty, largely achieved by piecing together pre-existing models\
(b) Lack of analysis of “zero-shot transfer” claims\
(c) Method working with a fixed number of pre-determined classes\
(d) Limited analysis of efficiency\
(e) Relatively small (2-3%) improvements\
(f) Inconsistent/unfair comparisons with the baselines

Some of these concerns were addressed well by the authors. For instance, applying semantic fusion and multi-view detection is a contribution of this work and (f) should not be a concern. (e) is addressed by clarifying that meaningful improvements on this benchmark depend on overcoming multiple independent challenges, which are not the focus of this work.

The AC still believes that some of the main concerns are not convincingly addressed. The AC agrees with most of the reviewers that there is not much novelty beyond integrating existing methods. The authors mention that lack of training is a mark of the power of the representation, however, that is not accurate since the failures of these methods propagate, and there is not much robustness or recovery. Regarding the real-world experiments, the authors mentioned “Demonstrations only show that the system is capable of doing something, it does not make a claim about generalizability or overall performance.” For top-tier conferences such as ICLR, a demo does not meet the bar. Quantitative results and analysis are required. Also, it is not convincing that the classes are pre-defined, and the AC did not find the argument in the rebuttal convincing. These issues along with the other issues mentioned by the reviewers preclude acceptance at this point.

**Reviewer Concerns:**

Please refer to the box above.

**Reviewer Scores:**

The reviewers criticized the novelty (5ipo, 9iMV, Yd22, JyoA, wuZW, and UXNY) would not change their score. Reviewer gr26 would lower the score since the concerns about pre-defined classes and real-world experiments are not addressed well.

---

### Decision · Program_Chairs · 2026-01-26

Reject